# Efficient pheromone navigation via antagonistic detectors in *Caenorhabditis elegans* male

Xuan Wan [1,2], Tingtao Zhou[3,4,6], Vladislav Susoy[5,6], Alessandro Groaz[1,2], Core Francisco Park[5], John F. Brady[3,4], Aravinthan D. T. Samuel [5] & Paul W. Sternberg [1,2] ✉

Chemotaxis to a moving potential mate that emits a volatile sex pheromone poses a navigation challenge requiring rapid, precise responses to maximize reproductive success. Volatile chemicals form gradients that differ from soluble compounds, potentially making navigation based on comparisons between spatially separated sensors unreliable for small-bodied animals. Here we show that, rather than a simple spatial comparison, *Caenorhabditis elegans* males employ an antagonistic strategy, comparing inputs from sex-shared head (AWA) and male-specific tail (PHD) sensory neurons with distinct response properties. Despite sharing a receptor, SRD-1, these detectors play different roles: AWAs promote forward movement and acceleration, while PHDs induce reversals and deceleration. In rising pheromone gradients, AWA activity dominates; in falling gradients, AWA inactivates, allowing PHD to correct trajectories. AWAs are essential for mate-searching, while PHDs are crucial for complex tasks. A minimal computational model reproduces these behaviors and infers how head–tail signals are combined. Thus, a sexually dimorphic, antagonistic sensory system enables adaptive navigation in dynamic environments

For most animals, finding a mate is an essential task and is an especially significant challenge for organisms that rely on volatile pheromones for mate detection, as the potential mate often moves in a complex environment and establishes a fleeting, low-concentration pheromone gradient. To obtain a mechanistic understanding of how animals can rise to this challenge, we studied navigation in *Caenorhabditis elegans*, as it has a highly accessible nervous system.

Self-fertilizing *C. elegans* hermaphrodites emit a volatile pheromone when sperm are depleted[1–5]. Volatile chemosensory signals are fundamental to facilitating complex behaviors such as strategic navigation, active sampling, and sequential decision-making across diverse species; they are crucial for reproductive success[6–8]. As sex

pheromones guide both innate and learned mate-searching activities in many animals, they provide an opportunity to study the neurobiological underpinnings of sexually dimorphic behaviors[9–18]. For example, the differential responses of males and females to identical stimuli, such as sex pheromones, raise questions about the sensory and neural circuit bases of sexual dimorphism in behavior[19–22].

*C. elegans* produces both soluble and volatile sex attractants. Soluble nematode-specific ascarosides are modulators of social/sexual behaviors and developmental regulation[23–41]. In contrast, this study focuses on a volatile, non-ascaroside sex pheromone (hereafter referred to as "the volatile sex pheromone") that remains chemically unidentified but plays a critical role in sexual communication by

[1]Division of Biology and Biological Engineering, California Institute of Technology, Pasadena, CA, USA. [2]Tianqiao and Chrissy Chen Institute for Neuroscience, California Institute of Technology, Pasadena, CA, USA. [3]Division of Engineering and Applied Science, California Institute of Technology, Pasadena, CA, USA. [4]Division of Chemistry and Chemical Engineering, California Institute of Technology, Pasadena, CA, USA. [5]Department of Physics and Center for Brain Science, Harvard University, Cambridge, MA, USA. [6]These authors contributed equally: Tingtao Zhou, Vladislav Susoy. ✉e-mail: pws@caltech.edu

signaling fertility and sexual maturity, as it can only be produced by hermaphrodites with no sperm due to depletion or dysfunction[3,5]. As hermaphrodites with no sperm can be considered females, we classify this pheromone as a female sex pheromone. The only known receptor for volatile sex pheromone is the G protein-coupled receptor (GPCR) SRD-1, which is expressed in the sex-shared amphid neuron AWA in males only, implying underlying sexually dimorphic sensory processing by that neuron[4].

The complexity of neural circuit questions makes studies in simple models such as *C. elegans* highly useful to rigorously define neural algorithms used by animals. The neural circuitry underlying chemoattraction in *C. elegans* has been mapped with increasing precision, suggesting that sensory neurons process environmental stimuli with remarkable reliability[42]. Besides of the sensitive sensory neurons, it also has a network of interneurons that play critical roles in processing sensory information and guiding behavioral responses that collectively facilitate navigation and avoidance behaviors: AIA amplifies olfactory signals, AIB controls reversals, AIY mediates response to gradients, AIZ modulates turns, and AVA initiates backward movement[43–51]. These circuits facilitate essential survival behaviors, employing mechanisms such as temporal integration[42], rapid desensitization[52], and the modulation of behavioral responses based on previous experiences and current conditions[43,47,53–64]. In short, the probabilistic nature of behavioral outcomes in *C. elegans* can be traced to the dynamic state of the neural network beyond sensory neurons, potentially explaining the variability in behavior outcomes from identical stimuli in the same individual[53,65].

Previous studies have shown that *C. elegans* employs sensory detectors on opposite ends of its body for rapid avoidance responses to aversive stimuli like SDS, H2O2, and glycerin[66–68], where quick response is prioritized over accuracy. In contrast, our study uncovers a volatile attractant chemotaxis mechanism in which dual detectors are instead used for target searching, a process that demands both rapid responses and high precision to successfully locate the target. Here, we describe a unique attractant dual-sampling algorithm that utilizes controlled expression of a single receptor to achieve sexually dimorphic chemosensory responses and an innate, adaptive navigation strategy for efficient volatile chemical chemotaxis.

In this study, we discover that the male tail-specific neuron, PHD, in addition to AWA, expresses SRD-1 and responds to the sex pheromone as well. We demonstrate that PHD is specifically required for navigation during challenging tasks. Using optogenetics, we compare the contributions of head and tail neurons to behavioral responses. To understand their interaction, we investigate the neural circuitry connecting them, and revisiting their response properties, we find that head and tail neurons detect distinct aspects of the pheromone gradient and concentration. Based on these findings, we propose a computational model to help elucidate navigation strategies.

## Result

### Volatile pheromone receptor SRD-1 is expressed in PHD, a male-specific neuron in the tail

Male *C. elegans* sense the volatile female sex pheromone via SRD-1[4,5], a GPCR. In hermaphrodites, it is expressed only in ASIs in the head region and PHAs in the tail region. Whereas in males, SRD-1 is expressed in the AWA and ASI sensory neurons in the head region. In males, SRD-1 was reported to be expressed in a pair of tail neurons[69,70], which we now have identified as the newly discovered pair of PHD neurons using several transcriptional reporters, based on their location, morphology, and marker gene expression (Fig. 1A showing an extrachromosomal array strain (*srd-1P*::mCherry); Figs. 2A and S1B showing an integrated cGAL-UAS strain (*srd-1P*::cGAL; UAS::GFP); and Fig. S1C showing a marker for Ray B neurons, namely a cGAL-UAS strain *pkd-2*p::cGAL; UAS::GFP; *srd-1P*::mcherry). We further verified SRD-1 expression in PHD neurons using a split cGAL driver with *lin-48d* and

*srd-1* 5′ transcriptional control regions, where *lin-48* is known to be expressed in PHD neurons[71]. The expression of SRD-1 in PHD neurons was further supported by recent male-specific neuronal transcriptomics data, which revealed heterogeneous SRD-1 expression within these neurons[72]. RNA-seq indicates *srd-1* is detected in ~40% of PHD neurons under standard conditions; however, our integrated *srd-1P*–driven cGAL with UAS::GFP reporter shows uniform PHD labeling (near 100%). Our approach also generated a PHD-specific driver for targeted manipulation of PHD neurons for subsequent experiments (See Fig. 2). In *C. elegans* males, AWA is required for chemotaxis towards the sex pheromone[4,5]. While previous studies have linked male tail-specific neuron PHD to various aspects of male behavior[71,73], their specific function remains unclear.

AWA neurons are known to respond to multiple volatile attractants[74–78], whereas PHA neurons are involved in repellent avoidance behaviors[66,75,79], and ASI mediates multiple sensory responses, including repulsion[80]. The sex-specific expression pattern of SRD-1 suggests that the differing behavioral responses to the sex pheromone —strong attraction in males and weak repulsion in hermaphrodites— may result from the sexually dimorphic expression of the chemoreceptor SRD-1. AIn addition, the dual-site expression of SRD-1 in both head and tail neurons suggests that males may integrate SRD-1-mediated sensory inputs from the anterior and posterior regions to navigate effectively toward the volatile pheromone.

### Male-specific PHD is activated by the volatile pheromone

We next tested if PHD neurons activate in response to the sex pheromone (Fig. 1C). We observed that PHD is activated by the sex pheromone from both *C. elegans fog-2* hermaphrodites and *C. remanei* females. To test if this response was mediated via SRD-1, we repeated the experiment in *srd-1(eh1)* loss-of-function mutant males. *srd-1* mutant males showed no PHD activation upon sex pheromone stimulation (Fig. 1C). Therefore, male-specific PHD neurons sense volatile sex pheromone via SRD-1. These findings, combined with the known role of PHD in mate-related behaviors[71], suggest its potential involvement in pheromone-guided navigation. Calcium imaging demonstrates that SRD-1 expression in both head (AWA) and tail (PHD) neurons is not merely anatomical but functionally mediates pheromone responses, providing direct evidence that males employ dual detectors to sense pheromone sources.

### PHD neurons enhance mate-searching efficiency in complex environments

To test the hypothesis that these spatially separated detectors for pheromone actually aid navigation, we used five types of chemoattraction assays designed to represent varying levels of difficulty males might encounter in natural conditions, comparing intact males to those with histamine inhibited-PHD neurons (Fig. 2). We achieved specific expression of the HisCl effector (a chemogenetic silencer) in PHD under the control of a split cGAL-UAS line: *lin-48dP*::NLS::cGAL (DBD); *srd-1P*::NLS::cGAL (AD) (Fig. 2A). Our experimental design included wild-type (*him-5*) males with/without histamine treatment and PHD-HisCl lines without histamine treatment as controls. Our data showed no significant differences between the three control groups in their response to sex pheromones under all test conditions (Fig. 2C–G). Thus, the observed effects reflect histamine-induced PHD silencing, rather than genetic background or histamine alone.

We first challenged PHD-inhibited males to locate a sex pheromone source placed at the same distance (3 cm) from the source, using two different diameters (6 and 8 cm) agar-filled Petri dishes (Fig. 2C, D). PHD-inhibited males refer to males expressing HisCl in PHD via split-cGAL (*lin-48dP*::NLS::cGAL[DBD]; *srd-1P*::NLS::cGAL[AD]) crossed to 15 × UAS::HisCl::SL2::GFP and exposed to 10 mM histamine (see "Methods"). All males located the source within 30 min. PHD-inhibited males performed as effectively as controls at both distances,

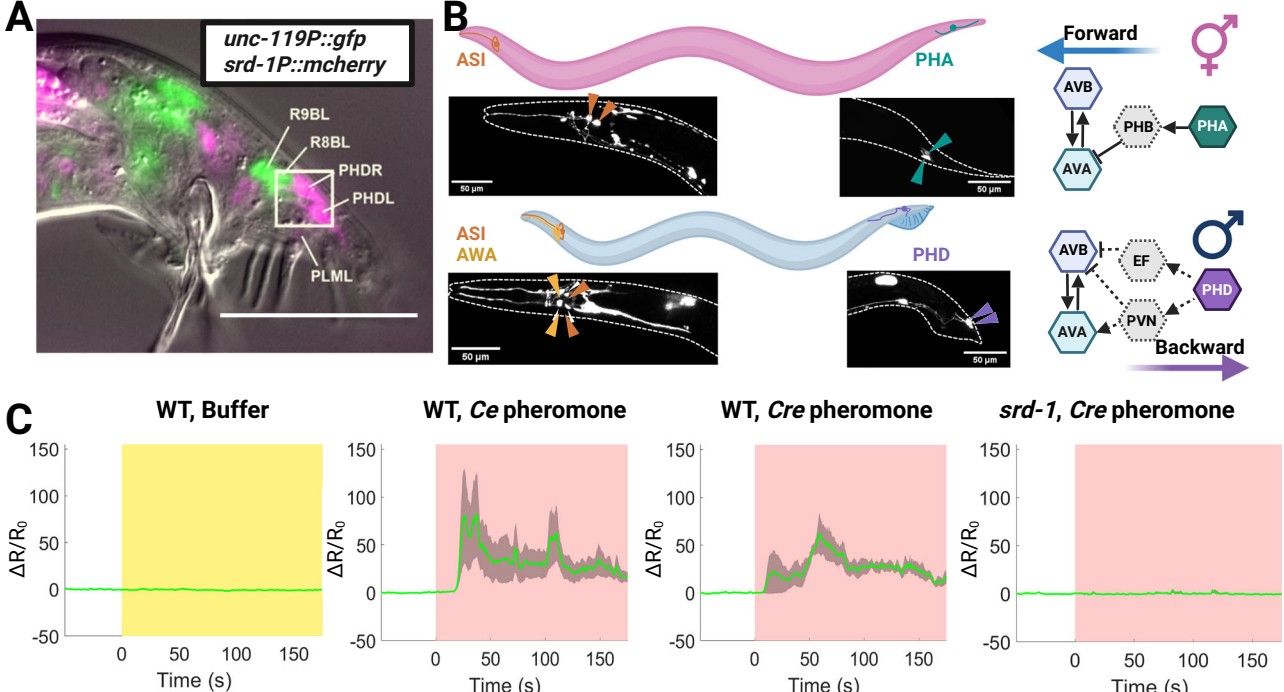

**Fig. 1 | *srd-1* sexual dimorphism in expression patterns and male PHD neuron excitation triggered by sex pheromone depends on the chemoreceptor SRD-1.** **A** Expression pattern of *srd-1*::mCherry (magenta pseudocolor) in the male tail. *srd-1* is expressed in a pair of PHD cells located in the tail region of male *C. elegans*. PHD neurons are defined based on their location in the tail ganglia, and its process morphology, with P*unc-119*::gfp indicating all neuronal cell bodies (green pseudocolor). SRD-1 is not expressed in Ray B neurons (Fig. S1) as reported in the literature before the existence of PHD was known[69,70]. Scale bar, 50 µm. **B** Sexual dimorphic expression of *srd-1* in both head and tail: *srd-1* GFP reporter expression in AWAs, ASIs, and male-specific neuron PHDs in males and in ASIs and PHAs in hermaphrodites. *rab-3P::mCherry* marks all neurons; an integrated cGAL-UAS line *(srd-1P::GAL4; UAS::GFP)* reports *srd-1* expression. The right panel shows the expression

pattern summary of *srd-1* in both males and females, along with the predicted neural circuits it might be involved in, based on their connectome and functional connectome insights. Scale bars as indicated. **C** Sex pheromone-triggered activation of PHD neurons, exhibiting SRD-1-dependent excitation, was observed through pan-neuronal calcium imaging with *rgef-1P*::GCaMP6s; a pan-neural marker (*rgef-1P*::mNeptune) indicated all neuron cell body locations (sample size: 5–7 worms per condition). Shaded areas represent the s.e.m. White indicates the pre-stimulation stage, and yellow shading indicates the time window of buffer application, while pink denotes the period of pheromone application. The green trace denotes the mean calcium ratio ($\Delta R/R_0$). The y-axis represents $\Delta R/R_0$, where R is defined as the fluorescence ratio of GCaMP to mNeptune. Created in BioRender. Wan, X. (2026) https://BioRender.com/9hzrztd. Source data are provided as a Source data file.

indicating that tail neuron input is not essential for simple, close-range navigation tasks. By contrast, when we increased the distance and placed males 5 cm from the source in an 8 cm-Petri dish (Fig. 2E), all test subgroup males were unable to locate the source.

We then tested males' ability to locate a more dilute odor. We diluted the sex pheromone concentration 10-fold and observed the responses of histamine-treated/untreated wild-type male (*him-5* male), histamine-treated/untreated PHD-specific driver with HisCl effector males (Figs. 2F, S2A). PHD-inhibited males exhibited notable difficulties in identifying the diluted pheromone source, unlike the three control group males. This differential response underscores the essential role of male tail PHD neurons in navigating toward weak pheromone signals.

To understand the effects of increased distance versus stimulus dilution, we conducted Finite Element Simulations of the pheromone concentration field (Fig. S3). In our assay setup, the stimulus was placed on the lid of the plate that was half-filled with agar, creating a bilayer distribution of the pheromone. As illustrated in Fig. S3A, the volatile component diffuses rapidly in the air, while the agar absorbs and then slowly releases the volatile component. The concentration field simulation revealed that in experiments where males were positioned far away from the pheromone source, they encountered areas of low absolute concentration without a clear gradient. Conversely, when males were tested nearer to a diluted pheromone source (3 cm), their performance, while reduced relative to controls, was better than in the 5 cm experiment, likely because the diluted source still

produced a weak but discernible Gaussian gradient despite its overall low concentration (Fig. S3B). These findings suggest that the concentration gradient plays a crucial role in guiding males to the pheromone source.

Since *C. elegans* navigate through three-dimensional (3D) soil and rotting fruit in their natural habitat, we examined the efficacy of PHD-inhibited males in a 3D search paradigm using gellan gum gel (see Methods), which likely better mimics natural conditions than a 2D Petri dish surface (Fig. 2G, H). In this endpoint assay, we recorded the number of worms that reach the target spot every 5 min. Worms must stop moving at the exact 3D location (depth included) of the pheromone to count as finding it. Control and test droplets are placed at the same depth to control for vertical bias, avoiding confounding from a preference for movement toward the surface or bottom. PHD-inhibited males had reduced proficiency in identifying pheromone sources within 3D spaces, underscoring the importance of PHD neuron inputs for environmental navigation that mirrors natural conditions (Figs. 2G, S2A).

Analysis of the distribution of males at multiple timepoints after the addition of stimulus revealed a uniformly low chemotaxis index for the PHD-inhibited male group across all intervals in weak signal and 3D exploration tasks (Fig. S2B). In our 2D experiments involving weak signals, most control males successfully located the target within 10 min. However, in 3D exploration tasks, the peak arrival time at the target extended to a significantly longer 20 min (Fig. S2C). Notably, we observed distinct impairments in the navigational behaviors of PHD-

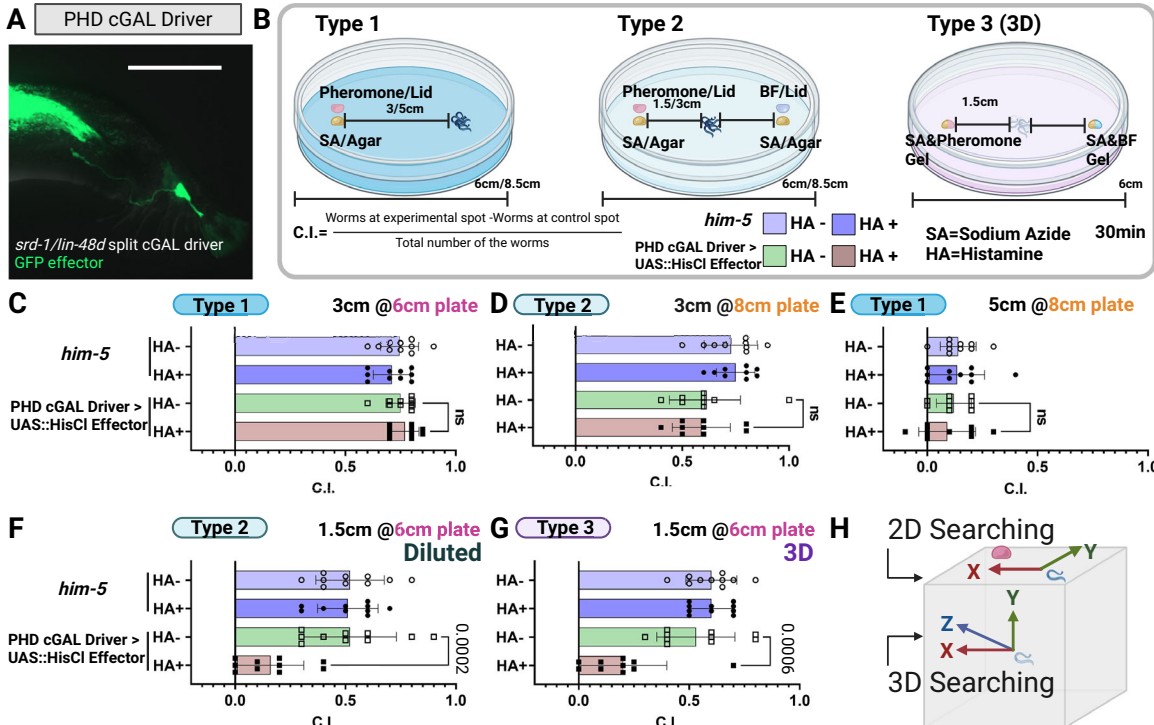

**Fig. 2 | Exploring the role of PHD neurons in mate-searching efficiency. A** Split cGAL construction of a PHD neurons-specific driver and validation of *srd-1* expression. Male tail fluorescence from *lin-48dP*::NLS::cGAL(DBD); *srd-1P*::NLS::cGAL(AD) crossed to 15 × UAS::GFP demonstrates GFP labeling of PHD, verifying *srd-1* expression and providing a driver for targeted PHD manipulation. Scale bar, 50 μm. **B** Schematic representation of the chemoattraction assay and the calculation of the Chemoattraction Index (C.I.). **C–G** Impact of PHD inhibition on male mate-searching efficacy under varied task complexities. Males expressing *lin-48dP*::NLS::cGAL (DBD); *srd-1P*::NLS::cGAL (AD) (driver) × UAS::HisCl (effector) were inhibited by histamine to silence PHD. Controls were the same genotype without histamine treatment, and WT *him-5* with/without histamine treatment, as indicated.

**C–E** Baseline chemotaxis: PHD-inhibited males locate sex pheromone comparably to controls in three tested conditions. **E** Sensitivity test (10× diluted pheromone): PHD inhibition significantly reduces detection and targeting performance. **G** 3D navigation (gellan gum/gel arena): PHD-inhibited males show impaired source finding relative to controls. The sample size for each assay consisted of 200 worms, 20 worms per trials, 10 trails. Not significant (ns): *P* > 0.05; exact *P* values are indicated on the figure. Two-tailed unpaired t-test. Error bars represent the s.e.m. **H** Schematic diagram of 2D (type 1 and 2) and 3D searching (type 3). Created in BioRender. Wan, X. (2026) https://BioRender.com/su7cm2k. Source data are provided as a Source data file.

inhibited males between weak signal detection and 3D spatial exploration tasks. In weak signal detection, the low chemotaxis index resulted from an equal distribution of males reaching both the test and control spots. Conversely, in 3D searching tasks, the low chemotaxis index stemmed from fewer males reaching both spots (Fig. S2B–D).

**Optogenetic dissection of SRD-1 positive neurons and PHD neuron's roles in male locomotion**

To examine the effects of head versus tail SRD-1 positive neurons on sex pheromone-induced behavior, we used the red-shifted Chrimson variant of channelrhodopsin[81]. We first evaluated the UAS::Chrimson effector driven with a pan-neuronal cGAL driver, demonstrating the system's efficacy (Fig. S4A): worms temporarily ceased movement during light stimulation, promptly resuming normal behavior once the light was switched off.

We then activated all SRD-1 positive neurons with Chrimson expressed under the *srd-1* regulatory control, using an *srd-1P* cGAL driver (Fig. 3A ATR- and S4C ATR+). The absence of ATR served as a control to negate any potential effects of light alone, confirming that the observed changes in speed were attributable from targeted neuron activation. Under ATR+ conditions (Fig. 3A), males exhibited continuous forward movement throughout the 12.5 s light stimulation period, transitioning to rapid steering or reduced speed after stimulation ceased, but not in ATR- males (Fig S4C).

To further confirm the role of SRD-1-positive neuron stimulation in modulating locomotor behaviors, we investigated the effects using extended stimulation protocols (30 s or 60 s light-on intervals

followed by 10 s light-off intervals) (Fig. 3C). During prolonged stimulation, forward movement was sustained, while turning behaviors remained consistently suppressed. Also, the cessation of stimulation was associated with reduced movement speed and an increase in turning rate, consistent with observed dynamics under shorter stimulation conditions.

Using an *odr-10P* cGAL driver crossed to a UAS::Chrimson effector, optogenetic activation of AWA in both sexes produced no detectable behavioral change, indicating that AWA activation alone does not drive forward locomotion or that *odr-10P* cGAL driver yielded insufficient Chrimson expression for activation (Fig. S4B).

We next targeted PHD neurons exclusively. Using a split cGAL driver under the control of *lin-48d* and *srd-1* promoters, we achieved PHD-specific Chrimson expression (Fig. 3B ATR+ and S4D ATR-). Under the ATR+ condition, PHD activation in males elicited a decline in average speed throughout the red-light stimulation period, after which males resumed movement. Under matched light stimulation, ATR− males exhibited no effect. This result contrasts with the overall movement patterns observed upon and after the activation of all SRD-1 positive neurons (See Fig. 3A). Previous studies also suggest that PHD neurons have a potential role in maintenance of backward locomotion during the mating partner body scanning[71]. We also observed that repeated activation of PHD neurons also resulted in a gradual reduction in speed (See Fig. 3B, right panel), suggesting a persistent slowdown effect on locomotor behavior after activation.

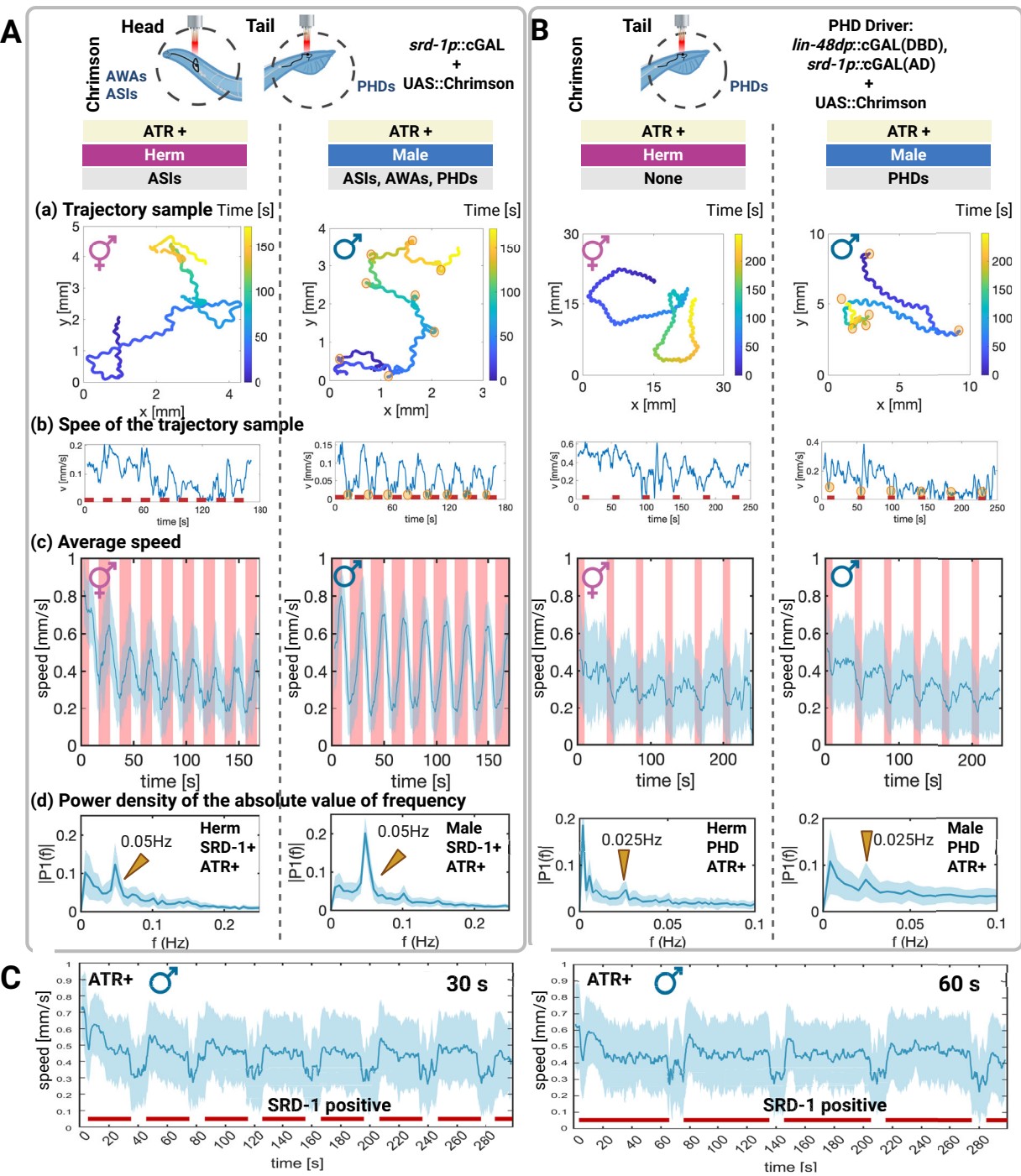

**Fig. 3 | Contrasting locomotor responses to optogenetics activation of SRD-1 positive neurons and PHD neurons.** Schematic of optogenetics with region-specific genetic targeting: (**A**) head and tail region AWA, ASI and PHD neurons (*srd-1P*−cGAL > UAS::Chrimson) versus (**B**) tail PHD neurons (split cGAL: *lin-48dP*::NLS::cGAL[DBD]; *srd-1P*::NLS::cGAL[AD] > UAS::Chrimson). **A** Sex-specific locomotor responses to SRD-1 positive neuron activation in *C. elegans*. SRD-1 positive neurons (*srd-1P*-NLS-cGAL driver) were activated in both sexes using Chrimson (UAS::Chrimson effector). The stimulation protocol consisted of 12.5-s light-on intervals followed by 7.5-s light-off intervals, forming a 20-s cycle. Light stimulation is represented by the pink bar in the figure. Sample size: about 120 worms. **B** PHD neurons were activated by expressing Chrimson via a split-cGAL driver (*lin-48dP*::NLS::cGAL[DBD]; *srd-1P*::NLS::cGAL[AD]) crossed to 15 × UAS::-Chrimson; animals were treated with or without ATR and stimulated with light during the assay. ATR− and light-off served as negative controls. The stimulation protocol consisted of 10-s light-on intervals followed by 30-s light-off intervals, forming a 40-s cycle. Light stimulation is represented by the pink bar in the figure.

The sample size for the SRD-1 positive neuron and PHD neuron activation assay included about 120 worms per sex. (See ATR- control in Fig. S4) (**a, b**) A representative trajectory sample and its corresponding time series of instantaneous speed magnitude, total speed V(t), are shown. **b** The red line indicates periods of light stimulation. The yellow circle labeled all the turning events. **c** V(t) averaged over 120 worms. **d** Fourier analysis of V(t) from Fig. 3A, B demonstrates that optogenetic activation of SRD-1 positive neurons and PHD neurons produces expected peaks at 0.05 Hz and 0.025 Hz, corresponding to the 20-s and 40-s stimulation intervals, respectively. **c, d** Shaded regions indicate ± s.d.; solid lines are means. **C** Persistent enhancement of forward movement upon activation of SRD-1 positive neurons, sustained for up to 60 s. The stimulation protocol involved 30- or 60-s light-on intervals followed by 10-s light-off intervals. Sample size: 100 (30 s stimulation) and 110 (60 s stimulation) worms. Pink/red bars indicate periods of light stimulation. Shaded regions indicate ± s.d.; solid lines are means. Created in BioRender. Wan, X. (2026) https://BioRender.com/lsxa7k1. Source data are provided as a Source data file.

Since we also observed behavioral responses to light stimulation in hermaphrodites, we performed a quantitative analysis to distinguish the robustness of these responses in hermaphrodites and males (Fig. 3). Hermaphrodite responses were more variable upon optogenetic activation, as also seen in the average speed (Fig. 3c) and Fourier analyses (Fig. 3d). Fourier analysis of total speed during SRD-1-positive PHD neuron stimulation revealed prominent peaks at 0.05 and 0.025 Hz, corresponding to the 20 and 40 s stimulation protocols, respectively (Fig. 3d). The amplitude and sharpness of the spectral peak at 0.05 Hz (or 0.025 Hz) quantifies the strength of the signal, i.e., the consistency and robustness of the behavioral response to periodic stimulation, and it is consistently stronger in the males' response than the hermaphrodites' by about a factor of two. Hence, we concluded that males display a more robust response. The more variable response in hermaphrodites also manifested as larger inter-individual differences, as shown by comparing the error bars (shaded area) in Fig. 3c in the left two panels. Both these analyses suggest a sex-specific effect of SRD-1-positive neuron activation. Upon PHD neuron stimulation, the male also showed a variable response.

## Head and tail neurons mediate distinct micro-behavioral responses to optogenetic activation

To confirm and further investigate how activation of SRD-1 positive neurons in the head versus the tail affects behavior, we optically targeted either the head or tail of freely moving males expressing *srd-1* Chrimson cGAL-UAS.

Initial observations showed that males predominantly moved forward in the absence of stimulation (ATR− controls; see Figs. 4, S4 and S5), so a short observation window could lead to a false positive result on forward movement. To evaluate the duration and persistence of this response, red light is applied to the head region for 30 s (Fig. 4A). As shown in Fig. 3, tail stimulation primarily induced reversals, a rapid transition from forward to backward movement. For the tail region stimulation, we chose short and repeated pulses alternating every 2 s over a 15-cycle period to capture multiple reversals (Fig. 4A).

After postprocessing, we divided the recording into 2-s segments. For each 2-s segment, we tabulated the behavioral motif displayed by the male or the first behavioral transition that occurred during that segment. We used two complementary behavioral classification frameworks to capture distinct aspects of locomotion dynamics. Four-Motif Analysis (Fig. S5) focuses on core locomotion states (forward, backward, forward-to-backward transition, backward-to-forward transition) to highlight macro-level movement changes during head/tail stimulation. Six-Motif Analysis (Fig. 4) expands categorization to include nuanced behaviors like self-exploratory (where the worm attempts to probe its own body) and high-frequency direction transitions (marked by frequent direction changes within the 2-s window), which better capture micro-behavioral variability.

A short light pulse delivered to the tail region induced a transition from forward to backward movement, accompanied by increased directional hesitation (Fig. 4B). During these hesitant periods, males frequently switched between forward and backward locomotion. However, we observed substantial individual variability in response to tail stimulation. While most worms reliably responded to light stimulation for 7 pulses, they responded an average of 4.3 times. A subset completely ignored the stimulus. This behavioral heterogeneity may arise from the variable expression of SRD-1 in PHD neurons[72].

With 30 s of head stimulation, we noted behaviors analogous to those from activating all SRD-1 positive neurons, including persistent forward movement during light exposure followed by rapid directional adjustment (Figs. 4C, S5C). We also observed that after head stimulation, males exhibited self-exploratory behavior (Fig. 4C), using their tails to probe against their bodies, a behavior that can persist for several minutes.

## Circuit that combines head and tail inputs to control locomotion direction

The male connectome[82] suggests locomotion command neurons are the likely locus of integration of the head and tail inputs. Specifically, PHDs have strong connections to PVN (ROI1) and EF (ROI2) interneurons[82] (See Fig. 1B), which connect to pre-motor command neurons. The direction of locomotion is controlled by reciprocal inhibition of forward and reverse command neurons[43,49,83,84]. To investigate the circuitry between sex pheromone sensory neurons and these motor command neurons, we specifically activated SRD-1 positive neurons with Chrimson and monitored pan-neuronal calcium dynamics using GCaMP6s (Fig. 5A–C, see Method). SRD-1 positive neurons, including PHD, AWA, and ASI neurons in males, had robust responses to light stimulation (Fig. 5C). In addition, two groups of neurons within the male tail region—presumably a pair of PVN neurons in one region of interest (ROI1) and a single EF neuron in ROI2 (Figs. 5B, C, S6)—had a strong positive correlation with PHD activity in ATR+ and not in ATR- males (Fig. S7). Here, we used normalized activity for regions of interest (ROIs) to facilitate comparison across populations of neurons with varying baseline fluorescence, while raw activity was analyzed for single neurons to preserve their intrinsic activity dynamics and avoid potential distortions from normalization. In the male head region, we observed a more negative correlation with light stimulation. About 4 neurons in the head region are positively correlated with AWA neuron activity, and about 10 neurons are negatively correlated (Fig. S7).

We observed a strong negative correlation between the activity of the reversal command neurons (AVA) and SRD-1-positive neurons in the head region (AWA and ASI). Although this inverse relationship was also present in the ATR-untreated control group, it was less pronounced. In ATR-untreated worms, the decrease in AVA activity may have been influenced by factors such as movement of the microscope objective or non-specific light responses.

## AWA and PHD have distinct response properties

It has been suggested that AWA neurons adjust their response based on fold-changes in odor concentration, reducing turning behavior with minimal increases in odor concentration[52]. We used prolonged, behaviorally relevant high concentration sex pheromone stimulation with a thin layer micro-diffusion (TLMD) unit (Fig. S8A, see "Method"), facilitating pan-neuronal calcium imaging (Fig. 5D, E)[4]. As shown in Fig. 5D, upon pheromone application, AWA and ASI neurons showed immediate calcium influx followed by rapid desensitization to a low, steady state within 1.5 min, maintaining this level for at least 5 min. This pattern indicates robust desensitization to even relatively high pheromone concentrations. PHD neurons exhibited sustained activation for at least 9.5 min with a gradual decline, indicative of their threshold-sensing properties (Fig. 5E). 62.5% of imaged males exhibited a rapid response to pheromone exposure and sustained high levels of activity, whereas the remainder displayed a slow response, with delayed activation (2.5 min post-stimulus) with a 40% reduced calcium signal amplitude (Fig. 5E, cropped to 250 s; D, full length 600 s). The fast and slow responses of PHD neurons are particularly intriguing. Recent male neuron transcriptomics data revealed a heterogenetic expression of SRD-1 in male PHD neurons, which may result in this individual variation[72]. This pattern may also be caused by PHDs being actively inhibited by other neurons. According to the connectome, PHDs are post-synaptic to several neurons[82], which may modulate their activity and contribute to the observed variability in response dynamics.

During the initial phase of neuronal activity, before the activation of PHD neurons, pheromone-stimulated calcium imaging revealed a negative correlation between reversal command neurons (AVA and AVE) and head sensory neuron activities (AWA and ASI) (Fig. 5D). This negative correlation suggests that the inhibition of reversal

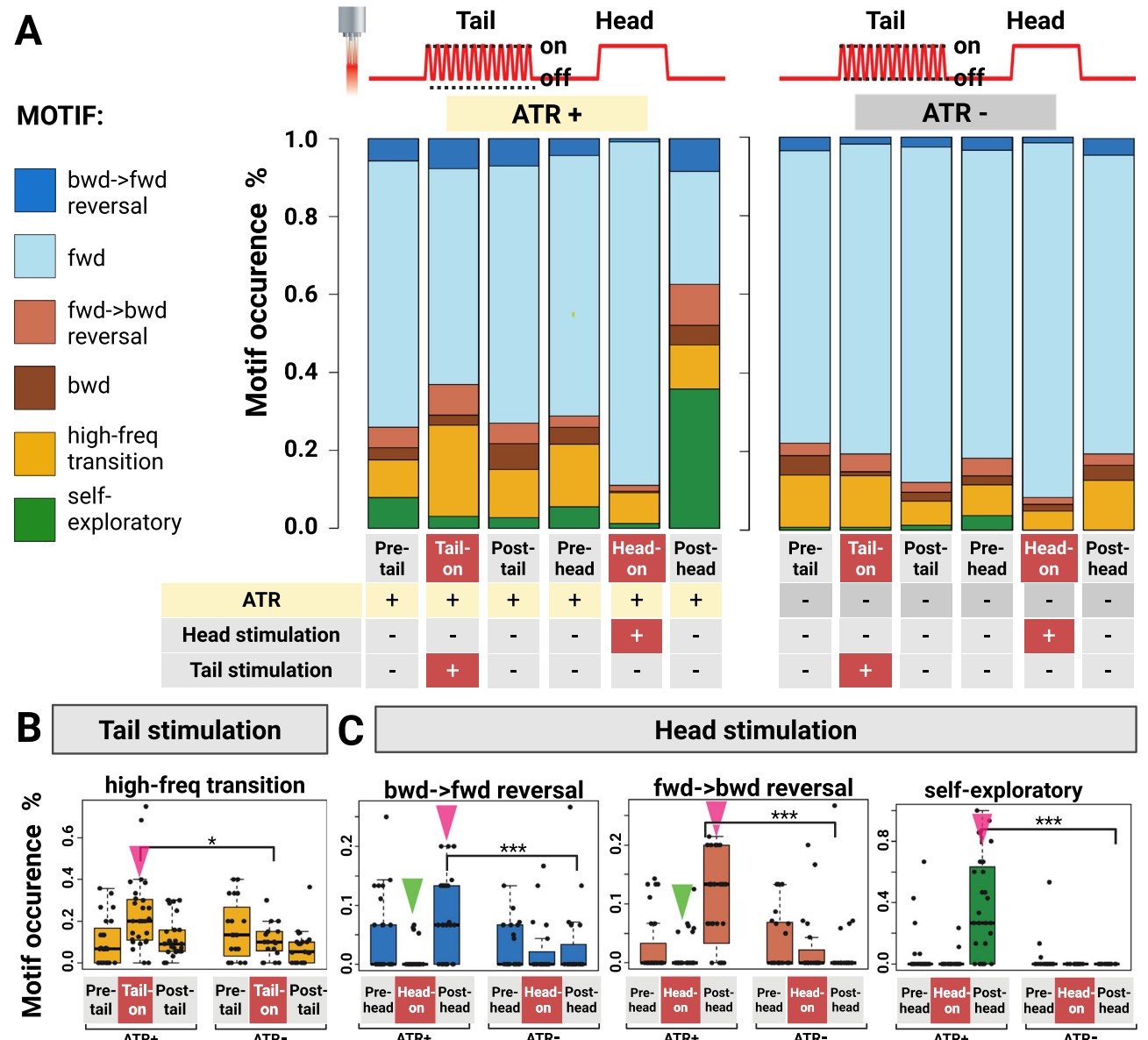

**Fig. 4 | Six-Motif Analysis: Characterizing micro-behavioral responses to optogenetic stimulation of SRD-1 positive neurons in male *C. elegans*' anterior and posterior regions. A** Summary of all characterized micro-behavioral and their portion among all behaviors. With *srd-1* drivers and Chrimson effectors in cGAL-UAS lines (*srd-1P*–cGAL > UAS::Chrimson) combined with region-specific light sources, we precisely targeted the head (AWAs and ASIs) or tail regions (PHDs) of freely moving males to analyze locomotion effects and characterize specific micro-behaviors. Behavioral segments were classified in 2-s windows as forward (fwd), backward (bwd), fwd-bwd reversal, bwd-fwd reversal, high-frequency direction transitions, or self-exploratory. High-frequency direction transitions, defined as >2 direction changes within a 2-s window, reflecting an exploratory strategy that cannot be unambiguously assigned to fwd-bwd or bwd-fwd. Self-exploratory, in which the worm engages in near-body probing behavior. **B** Tail-restricted illumination increases high-frequency transitions during light stimulation. All animals carried *srd-1P*–cGAL > 15×UAS::Chrimson and were reared ± all-trans-retinal (ATR). To probe tail-region circuits (PHDs), we delivered posterior-restricted light (illumination geometry and parameters in Methods). During light-on, ATR+ animals exhibited a robust increase in high-frequency direction transitions; this effect was absent in ATR− controls and in light-off trials. Behavior returned toward baseline after light-off. **C** Head-restricted illumination suppresses transitions during light stimulation and promotes transitions and self-exploration after light-off. Using the same *srd-1P*–cGAL > UAS::Chrimson genotype, we applied head-restricted light to engage head-region circuits (including AWAs/ASIs). In ATR+ animals, light-on suppressed direction transitions (both bwd-fwd and fwd-bwd). Following light-off, both transitions rebounded above baseline, and animals displayed increased self-exploratory "body-probing" behavior (tail pressed against/near the body). No significant changes were observed in ATR− or light-off controls. **A–C** The sample size for ATR+ group consisted of 18 worms. The sample size for ATR- group consisted of 12 worms. Green arrows indicate decreases in the ATR+ group during stimulation relative to pre-stimulation, post-stimulation, and ATR− group levels during stimulation. Pink arrows represent increases relative to the same conditions. ***P < 0.001; **P < 0.005; *P < 0.05. **B**, **C** Box plots indicate the median (center line); lower/upper box bounds are the 25th and 75th percentiles; whiskers extend to the minimum and maximum data points. Source data are provided as a Source data file.

command neurons by head neurons is pivotal for forward movement during gradient ascent. As demonstrated by the set of optogenetic experiments (see Figs. 3 and 4), gradient-sensing neurons in the head region dominate PHD input and primarily drive forward movement, suggesting that head neuron activity may override tail neuron activity.

In the subsequent phase of neuronal activity, AWA and ASI activity became hyperpolarized, and all PHD neurons were activated. This

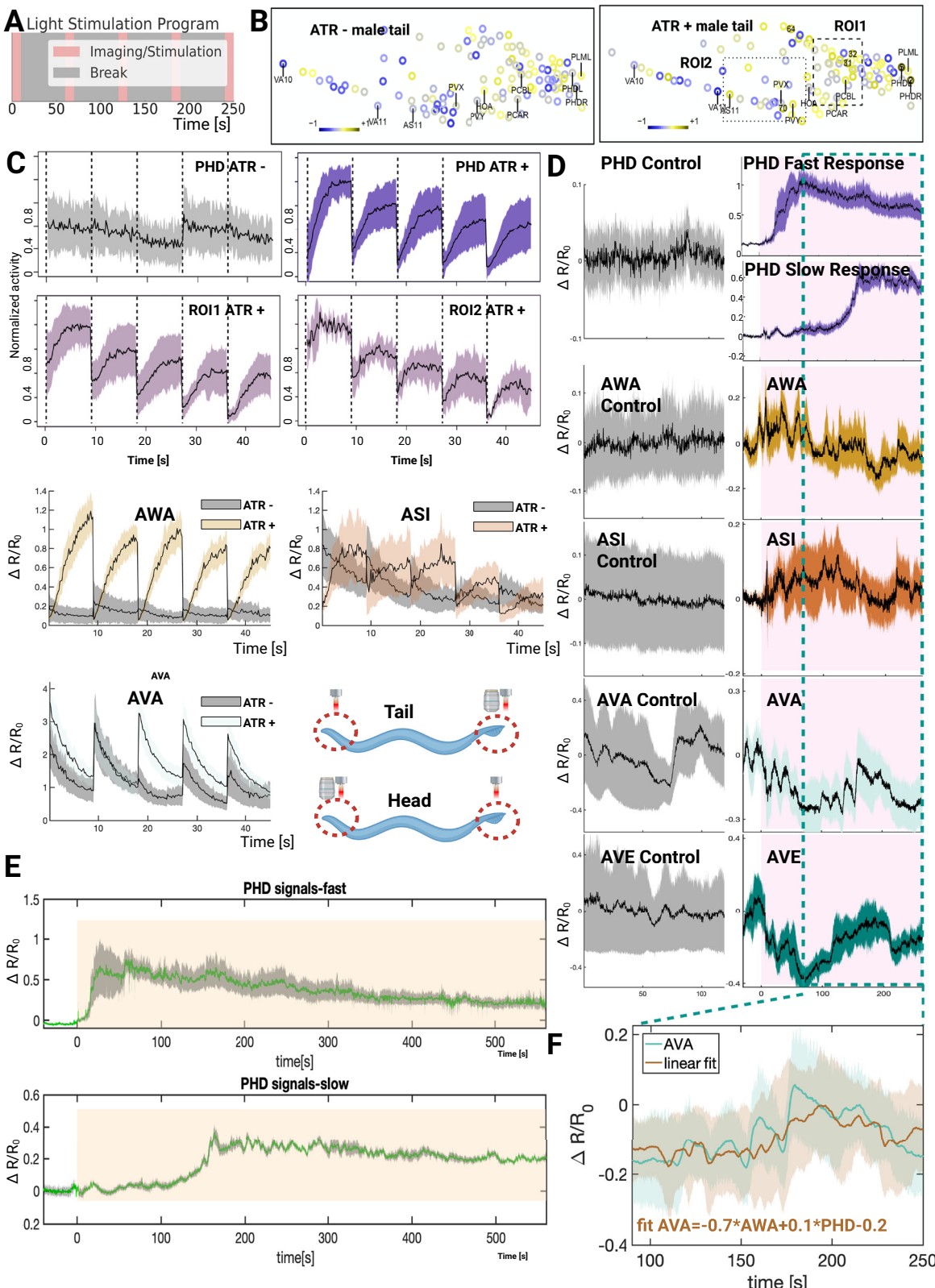

allowed us to examine the direct correlation between PHD neurons and reversal command neurons (AVA and AVE). We observed that reversal command neurons exhibited a negative correlation with head sensory neurons and a positive correlation with tail sensory neurons (PHD) (Fig. 5F), suggesting PHD-mediated activation of reversal command neurons, consistent with the observed reversal behavior during optogenetic activation (see Figs. 3 and 4).

## PHD and AWA neurons drive AVA activity at distinct concentration thresholds

To confirm the correlation between reversal command neurons and anterior and posterior SRD-1-positive neurons in different sex pheromone concentrations, and to determine the concentration threshold for behavior changes, we fabricated a head and tail multi-inlet microfluidic chip (Fig. 6A). We utilized immobilized worms within

**Fig. 5 | neural activation and correlation patterns following *srd-1* positive neuron optogenetic stimulation and pheromone exposure. A** Schematic representation of the optogenetic activation protocol, consisting of five 10-s imaging/stimulation intervals interspersed with 50-s breaks. **B, C** For optogenetic activation and calcium imaging, we expressed Chrimson by crossing an *srd-1P*::Gal4(sk)::VP64 driver to a 15 × UAS::Chrimson effector in the pan-neuronal imaging background. Animals were reared ± ATR, and defined light stimuli were delivered during imaging; Chrimson-dependent responses were observed only in ATR+ light-on trials. **B** Representative activation patterns were observed in male tail neurons. Through visual analysis, two regions, designated as ROI1 and ROI2, were identified as containing neurons with strong correlations to PHD activity (correlation coefficient >0.55) (See Figs. S6 and S7). The sample sizes: control (*n* = 9) and experimental (*n* = 12). **C** Traces of neuronal activities in PHD neurons and in two regions (ROI1 and ROI2), which often contain neurons strongly correlated with PHD. The sample sizes: control (*n* = 9) and experimental (*n* = 12) (See Figs. S6 and S7). Third and Forth row: neuronal activation in male head regions after

optogenetic activation of all SRD-1 positive neurons; sample sizes: control and experimental groups (*n* = 8 each, See Fig. S9). **D–F** Pheromone odor stimulation and calcium imaging. All panels use the pan-neuronal imaging strain and are exposed to volatile sex-pheromone extract; stimulus and the delivery method/protocol are detailed in "Methods". **D** Calcium responses in PHD, AWA, ASI, and reversal commanding neurons AVA and AVE upon 250 s of pheromone stimulation in both tail and head regions. Sample sizes for both regions are control (*n* = 5) and experimental (*n* = 8). **E** Calcium responses of PHD neurons over 9.5 min of pheromone stimulation, with data grouped by response initiation time into two categories. The sample sizes are 13 worms. **B, C, E** Shaded areas represent the s.e.m. **F** Fitting AVA neural activity data using a linear combination of AWA and PHD neural activity, focusing on the period both AWA and PHD neurons activated. This model indicates a weighted influence of AWA and PHD neuron activities on AVA neuronal response. Shaded regions indicate ± s.d.; solid lines are means. Created in BioRender. Wan, X. (2026) https://BioRender.com/lsxa7k1. Source data are provided as a Source data file.

microfluidic chips to precisely control the delivery of varying concentration stimulations to the head or tail independently. This setup enabled accurate stimulus application to distinct regions and facilitates switching between a control buffer and three different stimulus concentrations.

*C. elegans* has five pairs of premotor command interneurons, two of which promote forward locomotion and three of which promote reverse locomotion. These groups are interconnected and control distinct motor neurons[43,49,83,84]. AVA neuron activity peaks at the start of a reorientation[43,48,49,51,83] and AVA interneurons integrate opposing signals from head and tail sensory neurons, guiding behavior[67,85]. AVA's diverse activities across time scales position it as a key neuron in coordinating forward and backward movements[67,85]. Therefore, monitoring AVA neuron activity provides a useful indicator of the animal's forward and reversal behaviors.

Using diacetyl as a baseline attractant to validate our device (Fig. 6Bb), we observed that all tested concentrations led to decreased AVA neuron activity, suggesting an enhancement of forward movement. Diacetyl at higher concentrations, more effectively suppresses AVA neuron activity. We then tested pheromones at varying concentrations applied to both ends and separately to the head and tail (Fig. 6Bc). Simultaneous stimulus of both ends immediately decreased AVA activity, indicating forward movement. Remarkably, head-only stimulation caused a greater reduction in AVA activity compared with stimuli to both ends, whereas tail-only stimulation resulted in increased AVA activity (Fig. 6Bd,e); this difference suggests antagonism of head and tail input, consistent with our optogenetic and sex pheromone stimulation calcium imaging showing that head neurons activation promotes forward movement and tail PHD neuron activation triggers reversal (see Figs. 3, 4 and 6Ba). These increases and decreases in AVA activity only last for 1 min.

We performed area-under-curve (AUC) analysis on the AVA neural activity data (See Method). Notably, SRD-1 positive neurons in the head and tail show different preferences for sex pheromone concentrations. Higher pheromone concentrations more effectively suppress AVA activity when applied to both ends or exclusively to the head (Fig 6Bc–d). In contrast, AVA activation peaked at pheromone dilutions of 100–500X when applied only to the tail (Fig. 6C). But, at dilutions of 500X, 1000X, and 2000X, AVA neurons displayed predominantly random oscillations when pheromones were applied to the head or both ends (Fig. S10D). These findings suggest that higher pheromone concentrations promote forward movement more effectively, with head neurons strongly inhibiting AVA activity and weaker tail neuron activation contributing to sustained forward movement and increased speed. At low concentrations, the inhibition from head neurons diminishes, and tail neuron activation becomes more prominent, reducing forward movement. We also noted significant

spontaneous oscillations in AVA neuron signals across individuals, a finding shown in similar studies using microfluidic chips[65,86,87]. These fluctuations are likely due to the inherent spontaneous oscillation nature of AVA neurons, individual animals' internal status, and the restrictive environment of the microfluidic channels, which may discomfort the worms and trigger attempts to escape, causing variable AVA neuron signal patterns.

## Command neurons integrate antagonistic inputs from the head and tail

The integration of inputs from SRD-1-positive neurons in the head and tail regions reveals a sophisticated signal-processing algorithm that optimizes navigation. Optogenetic stimulation data (Figs. 3 and 4) demonstrate that activation of tail-region neurons induces reversal behavior and slows locomotion, while stimulation of head-region neurons suppresses turning, promotes forward locomotion, and increases speed. Notably, head-region activation alone mimics the effects of activating all SRD-1-positive neurons, highlighting the dominant role of anterior SRD-1-positive neurons in guiding behavior. This aligns with findings from AWA-defective *odr-7* mutants and AWA laser-ablated males[4,5], where AWA neurons are critical for locating pheromone sources, while ASI and PHD neurons are insufficient on their own. Conversely, the absence of tail-region PHD neurons impairs pheromone localization under natural, complex conditions, though it has minimal effects in ideal laboratory settings with high pheromone concentrations, reduced noise, and simpler navigation challenges (Fig. 2C, D). Together, these data suggest a dual-detector system in males integrating head and tail neurons' sensory input, with tail neuron activation acting as a corrective mechanism during navigational errors to enhance chemotaxis efficiency.

Pheromone stimulation calcium imaging data further elucidates this integration (Fig. 5D). By fitting AVA neuronal activity into a linear model incorporating AWA and PHD activity, the best-fit equation has AVA activity = −0.7 × AWA activity + 0.1 × PHD activity −0.2, indicating that AVA responses are primarily driven by inhibitory input from AWA neurons in the head, with a smaller excitatory contribution from PHD neurons in the tail (Fig. 5F).

Microfluidics chip-based experiments (Fig. 6D) further validated this integration mechanism. The area under the curve (AUC) of AVA neuron activation during head-specific and tail-specific stimulation closely matches the trends observed with dual stimulation. This suggests that the neuronal response results from a combination of antagonistic inputs from head and tail regions, providing a computational framework for navigating pheromone gradients. This dual-system integration reflects a sex-specific chemoreceptor-driven behavioral adaptation for efficient navigation in complex environments.

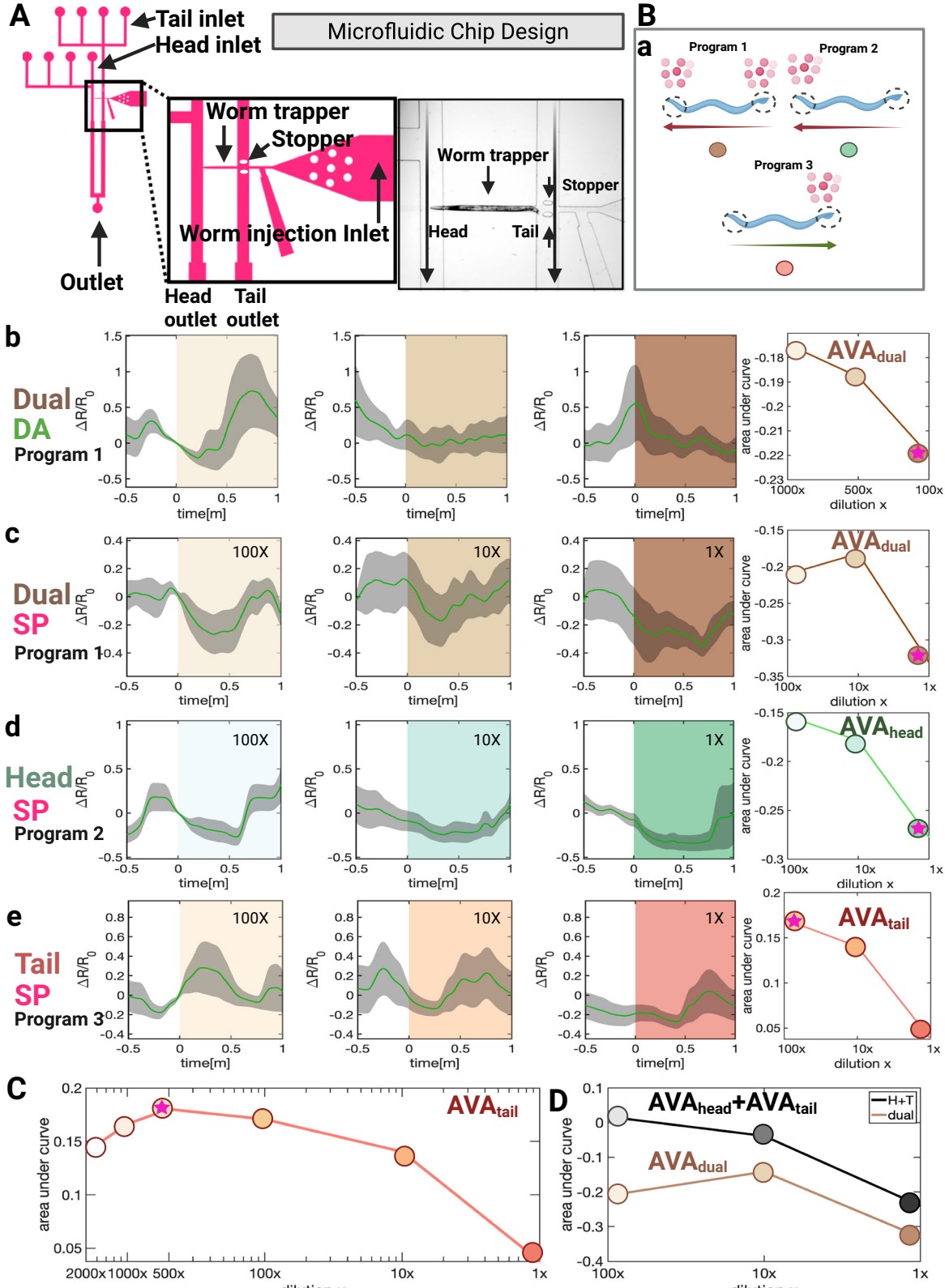

## A simple computational model for spatially separated detectors navigation in *C. elegans*

Sensory neuron inputs are typically conserved through the connectome and determine the behavioral outcomes[52]. Summarizing the experimental observations of the strong correlations between the AWA, PHD, and AVA neurons (Figs. 5 and 6), we propose a simple model to demonstrate the roles of head and tail sensory inputs. We

focused on macroscopic navigation behavior and neglected details of microscopic traits and changes in worm body curvature in motion.

*C. elegans* exhibits a remarkable ability to navigate complex environments using chemical cues. Two established directional navigation strategies have been observed[75,88–93]: (1) Klinokinesis, a random walk biased toward the stimulus, such as in bacterial chemotaxis, requires comparing concentrations over time to adjust random

**Fig. 6 | Dissecting reversal command neuron responses across different sex pheromone concentrations by microfluidics chip-based experiment.**
**A** Schematic of the head and tail microfluidic chip designed for precise, independent delivery of target concentration stimuli to the head and tail regions separately (See Method). **B** (a) Schematic diagram of stimulation program and predicted behavioral outcomes. **b**–**e** Effects of diacetyl and sex pheromone stimulation on AVA reversal command interneuron activity in *C. elegans* males. The calcium indicator GCaMP6s was used to monitor calcium dynamics in the AVA neuron. The AVA-specific split cGAL driver construct consisted of *gpa-14P*::NLS::cGAL (DBD)::gp41-1::N-intein::*let-858* 3′UTR and *rig-3P*::NLS::gp41-1::C-intein::cGAL(AD)::*let-858* 3′UTR, paired with the GCaMP effector construct 15 x UAS::GCaMP7b::SL2::mKate2. Diacetyl and dual-sided sex pheromone stimulation reduce AVA activity, as does head-specific pheromone stimulation, whereas tail-specific pheromone stimulation activates AVA neurons. Shaded areas represent the standard error of the mean (s.e.m.). The right panel shows area-under-curve (AUC)

analysis (see "Method"), indicating that dual-sided diacetyl and sex pheromone stimulation, as well as head-specific pheromone stimulation, exert the strongest inhibitory effects on AVA at the highest tested concentration. Tail-specific pheromone stimulation produces peak AVA activation at the lowest tested concentration (100-fold dilution). **C** Further testing of three additional diluted concentrations reveals that tail-specific stimulation elicits the highest activation of the AVA neuron at an intermediate concentration (500-fold dilution), with activation levels decreasing at both higher and lower concentrations. Each condition (head, tail, or dual location) at each concentration was tested with a sample size of 8 worms. **D** The area under the curve (AUC) for AVA neuron activation resulting from head-specific stimulation combined with tail-specific stimulation closely matches the trend observed with dual stimulation alone. The hot pink star indicates the concentration that resulted in the highest inhibitory or activating effect on AVA neurons. Created in BioRender. Wan, X. (2026) https://BioRender.com/5ttxkvk. Source data are provided as a Source data file.

turning that lacks directionality towards the correct heading. (2) Klinotaxis, a more refined navigation, utilizes directional turns to achieve more accurate gradient navigation[88,94–97]. Here, we modeled kinesis and taxis separately.

During kinesis searching, we modeled locomotion as a mixture of active Brownian motion[98] and run-and-tumble motion[99] (see "Methods"). Our first key assumption is that sensory inputs from the head and tail are converted into confidence variables $Q^H$ and $Q^T$, which then determine the worm's dynamical state (Fig. 7A). As the activity of reversal command neurons AVA and AVE is found to correlate negatively with AWA activation but positively with PHD neuron activation, we propose a simple form to qualitatively capture these correlations:

$$\gamma = \gamma_0 e^{-\lambda_K (Q^H - Q^T)}, \quad U = U_0 + U_1\left(Q^H - Q^T + \eta\right) \tag{1}$$

Here $\gamma_0 = 0.067/s$ is the baseline tumbling rate. $U_0 = 0.064$ mm/s is the baseline average speed, and $U_1 = 0.03$ mm/s is the magnitude of speed fluctuation. $\lambda_K$ represents the strength of klinokinesis, and $\eta$ represents a Gaussian noise in the speed. The worm's dynamic state is characterized by its speed and tumble rate $(U, \gamma)$. Loss of AWA or PHD input disrupts $Q^H$ or $Q^T$ computation, leading to impaired speed and turning regulation. The equations for $(U, \gamma)$ connect the nondirectional kinesis movements to confidence variables from head and tail inputs, which are subcircuit responses to stimuli on the head and tail sensory neurons. These mechanisms are also known as orthokinesis (for $U$) and klinokinesis (for $\gamma$).

Our model is based on the idea that sensory information coming from the worm's head and tail is translated into a signal that is a measure of 'confidence' ($Q$), and these signals directly determine the worm's behavior and movements (Fig. 7A). AWA neurons become more active as sex pheromone concentration increases[4]. In our microfluidics chip-based experiment, we found that input from head neurons suppressed AVA neuron activity (Figs. 5 and 6B-d). Therefore, in our model, the confidence signal based on head input ($Q^H$) will rise with increasing pheromone concentration until it reaches saturation (see blue line in Fig. 7A). Interestingly, high pheromone concentrations applied to the tail suppress AVA neuron activity, in contrast to the activation observed at low concentrations (Figs. 5 and 6B-e). Furthermore, continuous activation of the PHD neuron will result in a reduction in baseline speed and the sprint behavior exhibited upon proximity to the target, suggesting that $Q^T$ should decrease below zero to capture these observations (see red line in Fig. 7A). We propose using the $Q^H - Q^T$ model (see purple line in Fig. 7A) to represent the negative correlation between AVA/AVE command neurons and AWA activation, and the positive correlation with PHD neuron activation. Importantly, only $Q^H - Q^T$ accurately captures the observed increase in speed towards the end of navigation, which wouldn't be captured by

simply adding the two signals together (compare purple and green lines in Fig. 7A).

To determine the dynamics of $Q^H$ and $Q^T$ given external concentration field evolution, we adopt the simplest dynamics model with a single memory timescale:

$$\frac{dQ^H}{dt} = -k_1 Q^H + k_2^H f_H\left(C, \frac{dC}{dt}\right), \tag{2}$$

$$\frac{dQ^T}{dt} = -k_1 Q^T + k_2^T f_T(C)$$

Here $\frac{1}{k_1}$ is the timescale of memory, while the values of $k_2^{H,T}$ represent the sensitivity of SRD-1 positive neurons. $f_H$ and $f_T$ represent the sensory inputs, and they are chosen to qualitatively reflect the observations in this work and AWA calcium dynamics in[52] (see "Methods"). Key assumptions are: (1) the long decay time of PHD activity level during constant sex pheromone stimulation, as seen in Fig. 5F, indicates that PHD sensory input mainly depends on the absolute concentration. Hence, $f_T$ is only a function of $C$. (2) PHD response is positive at intermediate concentration ranges but becomes strongly suppressed at high concentrations, as shown in Fig. 6C. (3) The relatively short decay time of AWA activity level during constant stimulation, as shown in Fig. 2 of ref. 52, indicates that AWA sensory response mainly reflects the detected concentration change, with a mild increase as absolute concentration increases. For simplicity, in this model, worms will make directional (taxis) turns when they (1) detect a high enough concentration above a threshold value, and (2) if they currently experience a decrease in concentration. Such turns are assumed to be perfectly accurate toward the source direction. Based on our experimental observations, head-inhibited worms will not perform taxis turns.

We performed simulations for navigation on a small Petri dish with the concentration simulation for 30 min (Fig. 7B). Each simulation contains 10000 worms, initialized with random orientations. When both head and tail sensory inputs are missing, $Q^H = Q^T = 0$, the worms perform a standard active random walk with constant mean speed and tumble rate (Fig. 7C). Only a small fraction – about 10% of worms—were able to find the pheromone source by luck, as seen in the gray curve of Fig. 7D and Supplementary Movie 2.

Adding tail sensory input alone significantly enhanced the chance of finding the pheromone source, almost doubling the probability (Fig. 7D light blue curve). Adding head sensory input boosts the overall success rate to nearly 100%, simply due to the dominating effect of taxis turns (Supplementary Movie 1). Overall, with tail input, the trajectories are biased closer to the pheromone source, as shown in Figs. 7E and S11. In each simulation, worms start from a small region around the center of the plate, and the pheromone source is located at

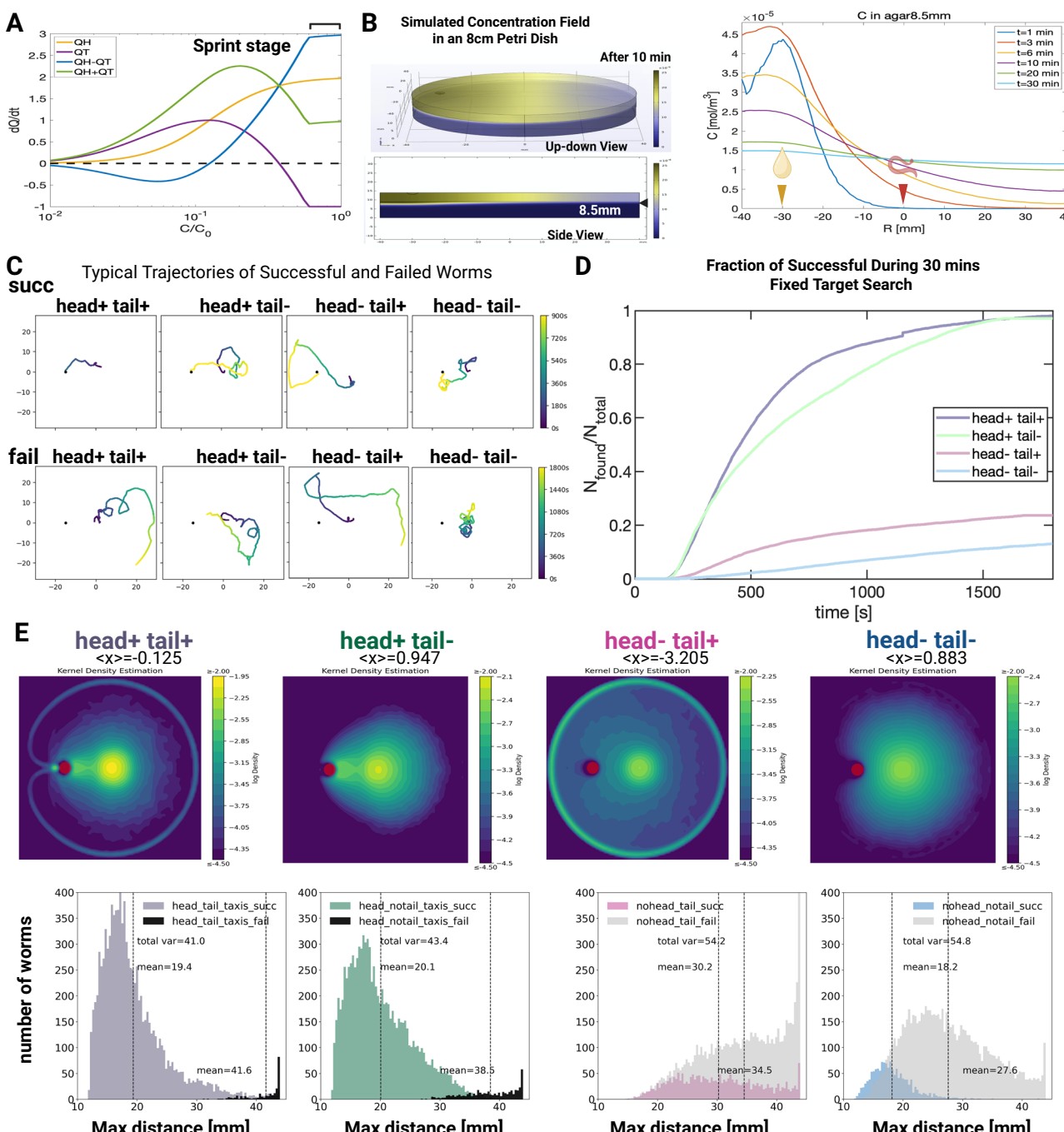

**Fig. 7 | Dynamic model for a navigation mechanism with antagonistic detectors. A** Schematic plot illustrating how confidence change rates depend on absolute concentration $C$ in the model. $C_0$ is the reference concentration at the pheromone source. For head input $Q^H$, a fixed concentration change rate is assumed for the sensory input. At low $C$, both head ($Q^H$) and tail ($Q^T$) inputs accumulate slowly. At intermediate $C$, tail input accumulates faster than head input. At higher $C$, tail input accumulates slower and is eventually suppressed, while head input signal is monotonically stronger as $C$ increases until saturation at very high $C$. The high $C$ regime corresponds to a large $Q^H - Q^T$ which leads to the sprint behavior. **B** Left panel: the simulated concentration field within an 8-cm Petri dish 10 min after stimulus introduction. The stimulus is on the lid of a plate half-filled with agar (9 mm thickness), establishing a dual-layer distribution for the volatile component. The black arrowhead marks the layers' interface in the bottom panel. Right panel: evolution of the concentration field at the air and agar interface (8.5 mm) over a 30-min duration. **C**–**E** Simulation results. **C** Typical trajectories of

successful and failed subgroups under different conditions of head/tail inhibition. **D** Fraction of worms that have found source during 30 min search task under each condition. Tail activation results in slightly better performance for head-activated worms (compare purple and green curves) and about twice better for head-inhibited worms (compare blue and gray curves). No taxis happen under head inhibition. **E** The upper panels show the simulated spatial distributions of worms averaged over 30 min. The red dot shows the location of the pheromone source, and all worms start at an average distance of 15 mm to the right side of the pheromone source. <x> is the mean position along the horizontal direction. The bottom panels show the histogram (*Y*-axis) of the maximum distance (*X*-axis) from the pheromone source that worms reached. Simulation sample size = 10,000 for each of the four different conditions. The mean of the maximum distance and total variance are indicated in each panel. Created in BioRender. Wan, X. (2026) https://BioRender.com/5ttxkvk. Source data are provided as a Source data file.

the red point on the left side. We denote the direction from the pheromone source to the plate center as the x direction and measured the average x position of all worms during the entire simulation. This is denoted as <x> on top of each panel in Fig. 7E. We notice that <x> is negative whenever tail input is present but becomes positive when tail input is absent. Hence, tail input enables worms to move on average to the left side of the plate, i.e., they behave more attracted to the pheromone source with tail inputs. We compute the maximum distance from the pheromone source that reaches worm reaches during the search, denoted as $d_{max}$. For successful worms with head inputs, the addition of tail input accelerates their search so that their $d_{max}$ is slightly smaller compared with those without tails. In the absence of either head or tail inputs, worms only perform random walks and stay near the starting position; adding tail input effectively incentivizes them to explore larger areas and be more biased to the source, manifested in significantly larger $d_{max}$ values. Overall, these simulations suggest that tail input significantly improves the success rate of mate-searching.

## Discussion

### Computational modeling suggests tail input suppresses mistakes

Building on our computational modeling of pheromone diffusion, we find that tail input is essential for suppressing navigation errors arising from the rapid diffusion of volatile attractants. Specifically, our finite element method (FEM) simulations reveal that the pheromone concentration field deviates from Gaussian distributions, and based on our simulations of the dynamical model, the resulting pheromone distributions can lead to artificial increases in head confidence $Q^H$ for worms in low concentration regions due to the finite plate size, leading to erroneous behaviors without tail input. Compared to water-soluble chemicals studied previously, air-borne pheromones diffuse rapidly, creating a nonlinear process where the agar initially absorbs pheromones to a high concentration before emitting them back into the air. This non-Gaussian behavior is characteristic of chemicals transported across air and agar phases and must be considered in chemotaxis models.

The errors caused by an artificial increase in head confidence can be mitigated by the tail. Tail input corrects these errors by increasing the tumble rate, ensuring more efficient directional adjustments. Moreover, suppressing PHD activity at high concentrations $Q^T \to -1$ significantly improves navigation, supported by GCaMP data showing reduced PHD activity under 1x concentration stimulation (Fig. 4). Consequently, we adopted a functional form for PHD response: $\lim_{C \to 0} Q^T \to 0$, $\lim_{C \to C_0} Q^T \to 1$, and $\lim_{C \to \infty} Q^T \to -1$. These dynamic properties suggest that the interplay between head and tail neurons is critical for robust navigation strategies, offering an extensible framework for modeling chemotaxis in complex environments.

Critically, our simulations also revealed that the concentration differences between anterior and posterior positions are negligible due to the small size of the worms. This fundamental constraint necessitates an alternative navigation strategy: rather than measuring spatial differences along the body axis, C. elegans employs antagonistic comparison between functionally specialized head (AWA) and tail (PHD) neurons. This computational innovation allows robust navigation where traditional gradient sensing would fail.

### Sexually dimorphic decision-making in C. elegans stems from sensory neuron inputs

The distinct responses of males and females to the same sex pheromone prompt questions about the neural foundations of behavioral sexual dimorphism[3–5]. Our findings indicate that sexual dimorphism in naïve C. elegans males' response to sex pheromone stimulation stems from variations in receptor expression. This leads to differential sensory neuron activity propagating through neural circuits, culminating in distinct behaviors. Although existing data from other organisms suggest that sex-specific pheromone responses arise from sexually dimorphic neural circuits rather than differential sensory perception[9,100,101], incomplete characterizations of neural networks in more complex model organisms may bias our interpretation of this phenomenon. C. elegans exemplifies an olfactory system where sophisticated modulation of a single receptor's expression allows both efficient chemotaxis and the execution of distinct sex-specific behaviors. Therefore, the behavioral outcomes in naïve animals during navigation can be predicted from the activity of the key sex pheromone sensory neurons.

PHD neurons remained active for ≥9.5 min with a slow decay (Fig. 5E). Across animals, responses segregated into two classes: fast responders (62.5% of males) showed an immediate, sustained increase at pheromone onset, whereas slow responders (37.5%) exhibited a ~2.5-min delay and ~40% lower peak ΔF/F. Two non-exclusive mechanisms could underlie this heterogeneity. (1) Molecular variability: single-cell datasets report variation in srd-1 abundance in PHD. Although both of our cGAL–UAS reporter lines labeled PHD uniformly (~100%), cGAL–UAS amplification can mask modest expression differences; thus, the heterogeneity is likely quantitative (more vs. less SRD-1) rather than all-or-none. Moreover, single-cell/single-nucleus RNA-seq can under-detect low or intermittent transcripts due to limited capture efficiency and technical dropouts/zero inflation—issues known to disproportionately affect sparsely expressed GPCRs[102–106]. (2) Circuit gating: PHD receives postsynaptic inputs from multiple neurons; state-dependent inhibition from these partners could delay onset and reduce amplitude in some animals. We therefore interpret the fast/slow classes as reflecting threshold-sensing in PHD modulated by receptor dosage and/or inhibitory factors. Discriminating between these mechanisms will require targeted experiments (e.g., srd-1 dosage manipulations and selective silencing of candidate presynaptic partners during PHD imaging).

### Decision in uncertainty: how worms navigate complex gradients with imperfect sensory input and inaccurate steering angle control

As a complementary approach to experimental observations of the worms' turning behaviors, we conducted a mathematical analysis for the optimal turn angle and compared it for consistency with our observed data. We modeled an idealized worm with precise short-term memory of its trajectory and the ability to precisely detect concentration gradients in its path by traveling along the same direction for a short duration. We assume that the ideal worm's sensory systems are infallible, and the worm's neural circuitry can process this information, such that sampling along two distinct directions will be theoretically sufficient to identify the gradient's direction. In practice, however, a non-ideal worm's memory and sensory perception are inaccurate due to inherent noise, complicating the task of gradient estimation. This challenge mirrors a regression problem where the objective is to select an optimal turning angle for a second sample that minimizes the uncertainty in determining the gradient's direction. We assumed a steady concentration field that does not vary with time during the sampling of two directions of a non-ideal worm.

Our analysis for a non-ideal worm, grounded in a least-squares-error regression framework, suggests that the optimal choice for a second sampling direction consistently exceeds 90° regardless of the initial sampling orientation relative to the actual gradient direction. With an initial orientation distributed uniformly randomly, the preferable turning angle is approximately 120° (Fig. S12). This direction is chosen to be away from the gradient if the initial angle is qualitatively aligned (less than 90° from the gradient) and towards the gradient otherwise. Given the worm's lack of precise knowledge of gradient direction, it faces a choice: turn 120° in either direction or, as a compromise, nearly 180° (Fig. S12). Observations suggest that worms likely

opt for turns close to 180° in early search stages, indicating a strategic approach to maximize information gain about the gradient's direction. Worms thus intricately balance their behavioral strategies with their sensation, motor control, and cognitive limitations.

## A navigation strategy based on an integrative dual-detector mechanism

Dual detectors to control directional movement allow simultaneous comparison of stimulus rather than a single organ comparison in time. In *C. elegans*, the phasmid neurons in the tail, similar to the amphid chemosensory organs in the head, perform a chemosensory function[66]. Indeed, amphid and phasmid neurons influence the worm's backward movements when exposed to repellents, integrating sensory inputs from both the head and tail to promote effective escape behaviors[66,68,107]. The AVA interneurons consolidate signals from the head and tail sensory neurons to steer behavioral choices[107]. Our study provides evidence of dual detectors in attractant navigation. Repellents demand rapid avoidance (reversal), while attractant sensing requires complex, targeted navigation strategies.

For repellent avoidance behaviors, worms employ a simple strategy in which both head and tail inputs trigger reversal[66,68,107], distinguishing it from the more complex mechanisms underlying attractant navigation. In attractant navigation (Fig. 8), head- and tail-located neurons utilize the same receptor but drive distinct and opposing sensorimotor responses: (1) AWA promotes forward movement and speeding up, while PHD induces reverse movement and slowing down. (2) Behavioral output is typically dominated by AWA activation, while PHD input provides antagonistic modulation of AWA-driven behavior. (3) The head neuron exhibits gradient-sensing properties and rapid desensitization in constant concentrations, while the tail neuron displays threshold-sensing properties and remains activated under constant concentrations. (4) PHD neurons are activated at lower pheromone concentration, thus preventing males from making false taxis turns induced by the rapid temporal evolution of volatile pheromone concentration. PHD neurons are inactive at high pheromone concentration, promoting rapid acceleration (sprints) as the male approaches the pheromone source. We computationally modeled the response, proposing that males have a measure of confidence that they are proceeding in the right direction; the absence of PHD input disrupts directional confidence, preventing the acceleration response. This demonstrates a fundamental aspect of neural circuitry: different neurons have evolved to handle distinct environmental stimuli.

Contrary to prior hypotheses[108–111], *C. elegans* does not rely solely on simple head-tail concentration comparisons but instead employs a more complex antagonistic mechanism involving head and tail sensory neurons, each with distinct preferred concentrations for optimal activation. Our proposed mechanism focuses not on the spatial separation of receptors but on comparing signals from two distinct sensory neuron types, AWA and PHD, with different response properties. This comparison enables the animal to integrate various aspects of pheromone gradient dynamics and make adaptive behavioral decisions. While the mechanism relies on comparing outputs from these neurons, their differences stem more from their response characteristics rather than their physical locations. This strategy would be particularly beneficial for navigating a volatile gradient, such as a shallow and low-concentration pheromone gradient, where there is minimal difference in concentration between the worm's head and tail, making head-tail absolute concentration comparisons very unreliable.

To assess whether spatial separation between AWA (head) and PHD (tail) produces a concentration disparity in what they sense, we quantified the relative difference $|C(\text{head}) - C(\text{tail})| / C(\text{body})$ ($C$, concentration) across our concentration field simulated arenas (Fig.

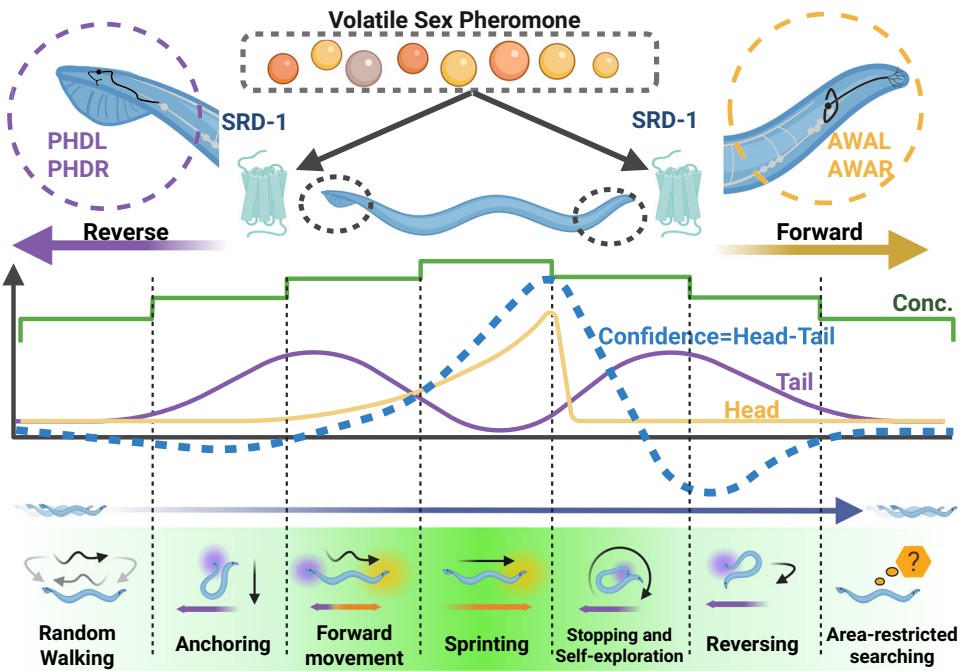

**Fig. 8 | Spatially separated pheromone detection drives adaptive navigation in *C. elegans* males.** The **upper panel** illustrates how *C. elegans* males utilize spatially separated pairs of sensory neurons, AWA (head) and PHD (tail), that detect the same volatile sex pheromone molecule via the receptor SRD-1. Despite sharing this receptor, head neurons detect gradient changes and activate when concentration increases, suppressing turning, and promoting forward movement and acceleration. In contrast, tail neurons detect absolute concentration and activate at preferred moderate concentrations, inducing reversals and deceleration. The **bottom panel** shows that upon initial pheromone introduction, simultaneous activation of head and tail neurons cancels each other out, causing males to transition from random walking to a stationary anchoring state, preventing them from chasing the concentration wave initially. As pheromone concentration increases, head neurons dominate, driving forward locomotion and acceleration. At the highest concentrations, tail neurons become less active while head neurons are highly activated, leading to rapid, goal-directed movement (sprinting). Incorrect directional choices result in AWA deactivation and persistent PHD activity, inducing pausing and local exploration. Repeated unsuccessful attempts trigger area-restricted searching. Created in BioRender. Wan, X. (2026) https://BioRender.com/5ttxkvk.

S3C). Specifically, we computed this metric and then averaged over (1) the full plate area, (2) the entire simulation duration, and (3) all worm orientations $\theta \in [0, 2\pi]$. The resulting empirical distribution is strongly concentrated near zero (probability density shown on a log scale), with a 99th percentile of 0.1056. Thus, for >99% of space-time-orientation, $|C\,(\text{head}) - C\,(\text{tail})| / C\,(\text{body}) < 0.1056$. The rare larger values arise predominantly near plate edges, where $C$ is itself very small, and the ratio becomes numerically noisy, rather than reflecting meaningful signal gradients.

Consistent with this result, our model already incorporates stochasticity in head and tail confidence. Across relative noise magnitudes of 0.01–0.1, model outputs changed minimally. Because the typical head–tail concentration contrast is of the same order or smaller than this noise amplitude, we do not expect it to carry reliable information for navigation under our conditions. These quantitative results justify our simplifying assumption to ignore explicit head–tail concentration differences in the main model without altering the qualitative conclusions.

Besides our work, other studies have shown that numerous chemosensory GPCRs are also co-expressed in the head amphid and tail phasmid neuron classes[66,112–116], which raises the possibility that this antagonistic detector mechanism may represent a general strategy in *C. elegans* navigation. This mechanism, integrating distinct neuronal responses to environmental stimuli, allows precise adaptation to dynamic and complex sensory landscapes. Beyond pheromone navigation, the principles uncovered here likely extend to other volatile chemosensory contexts, providing a broader framework for understanding how animals solve the universal challenge of searching for moving targets within noisy and variable environments.

## Methods

### Worm strains
Strains of *C. elegans* used in this study originated from the N2 (Bristol) wild-type lineage and were maintained under standard laboratory conditions using Nematode Growth Medium (NGM) plates with *Escherichia coli* OP50 as the food source. Detailed information regarding the strains is provided in Supplementary Table 1.

### Statistics and reproducibility
All experiments were performed on *C. elegans* adults, with individual worms treated as independent biological replicates. Sample sizes were chosen based on prior work in the field, pilot experiments, and practical considerations (e.g., expected behavioral variability and assay throughput), and are reported in the figure legends and/or Methods for each experiment. No statistical method was used to predetermine sample size. For tracking-based optogenetics assays, trajectories were excluded using pre-established quality-control criteria: worms were excluded if they physically contacted another worm during the assay or reached the plate edge; across ~150 imaged worms per sex, ~120 trajectories passed quality control and were analyzed. Outside of these pre-defined criteria, data exclusion is stated explicitly where applicable. Statistical analyses (including curve fitting and linear regression where used) were performed using custom analysis code, and the specific statistical tests, exact n, definition of center and error bars/box plots, and significance thresholds are provided in the corresponding figure legends. The behavior classification was randomized. The Investigators were blinded during outcome assessment.

### Microscopy and imaging
Animals were immobilized on a 4% ultrapure agarose pad using 3-5 μL of 2–5 mM levamisole. The imaging was conducted using an inverted laser scanning confocal microscope, Zeiss LSM 980, integrated with a Zeiss Axio Observer 7 with Definite Focus 2. Subsequent image processing was executed via FiJi (ImageJ) software.

### Inhibition of PHD neurons by the HisCl method and modified chemoattraction assay
Inhibition of PHD neurons was achieved through the histamine-gated chloride channel method, as outlined in the study[117], with slight alterations. Nematode Growth Medium containing histamine (NGM-HA) was prepared by incorporating histamine dihydrochloride from Sigma Aldrich (1 M stock solution) into NGM agar at approximately 65 °C just prior to pouring the plate. Control plates without histamine were prepared simultaneously from the same batch of NGM. Plates supplemented with 10 mM histamine remained effective for a minimum of 2 weeks when refrigerated at 4 °C, verified by preliminary testing with a control strain possessing the *myo-2* promoter-driven HisCl effector in the cGAL-UAS system. L4-stage males were selected and isolated a day in advance of the assays. Histamine treatments were conducted without the presence of food to eliminate confounding factors. Histamine-induced inhibition occurred within 10 min of exposure. Once removed from histamine, the animals typically require 1–2 h to fully recover. For PHD-specific inhibition, we used the split-cGAL driver (*lin-48dP; srd-1P*) crossed to 15 × UAS::HisCl::SL2::GFP; animals on 10 mM histamine (NGM-HA) are referred to as "PHD-inhibited," whereas histamine-free condition worms served as controls

The chemoattraction assay plates were prepared using 1.5% agar, 25 mM NaCl, 1.5 mM Tris-base, and 3.5 mM Tris-Cl, and conducted according to the protocol described in the literature[4], but included modifications to accommodate different plate dimensions, adjustments in the spatial distance between the stimulus source and the starting point of the animals, and alterations in the sex pheromone concentration, as depicted in the associated figures (see Figs. 1 and S1).

Stimulus droplets are placed on the lid of the plate, and the location of the worms is recorded every 5 min up to 30 min post-stimulation to assess their response to the introduced stimulus. In our research, we utilized two sizes of plates for the assays, with effective worm-moving surfaces measuring 5.5 and 8.4 cm in diameter. For simplicity and clarity in labeling, these were referred to as 6-cm and 8-cm plates, respectively. In our study, we employed two types of 2D chemoattraction assays distinguished by the setup and calculation of the Chemoattraction Index (C.I.). For Type 1 assays, constrained by the smaller plate size, a control buffer spot was not included. Here, the C.I. was derived from the ratio of worms that reached the test spot to the total number of worms introduced. Conversely, in Type 2 assays, the C.I. was determined by subtracting the number of worms that reached the control spot from those that arrived at the test spot, divided by the total number of worms. In each chemoattraction assay, we placed 20 worms at a designated starting point. To ensure the reliability and robustness of our data, we repeated each assay 20 times across three distinct days.

### Chemoattraction assay
Male chemotaxis was quantified using a chemoattraction assay as previously described (Wan et al.). Briefly, 2 μL of pheromone extract or M9 buffer (control) was applied to designated spots on the assay plate, each supplemented with 1 M sodium azide to immobilize responding males. M9 buffer (3 g KH$_2$PO$_4$, 6 g Na$_2$HPO$_4$, 5 g NaCl, 1 mL 1 M MgSO$_4$, water to 1 L; autoclave). 20 one-day-old adult males were placed at the origin point, and the chemoattraction index (C.I.) was calculated after 30 min using the formula: C.I. = (E − C) / (E + C + N), where E, C, and N represent the number of worms at the experimental, control, or neither spot, respectively. PHD-inhibited worms were prepared via the "Inhibition of PHD Neurons by HisCl Method".

3D chemoattraction assay employing 0.3 g of gellan gum in 100 ml chemoattraction assay buffer to create a navigable gel environment. This composition ensures that worms can navigate through the gel freely, without becoming immobilized by water or causing structural damage to the gel through movement. The gellan gum

solution is adjusted to match the salt concentration of the chemoattraction assay plates utilized in this research.

For plate preparation, 22 ml of the gellan gum solution is poured into a 6-cm plate immediately after the gellan gum is dissolved by heating in a microwave. The solution is allowed to cool to room temperature before introducing the worms. To embed the worms within the gel, 0.5 ml of the gellan gum gel is removed from the center of the plate by pipetting. Then, 20 worms are picked by a worm picker and placed. This cavity is then refilled with 0.7 ml of gellan gum gel, ensuring the worms are fully embedded within the gel. Unlike traditional chemoattraction assays, the stimulus is applied directly onto the surface of the gel rather than on the lid of the plate (see Fig. 2B). This approach ensures that the pheromone remains accessible to the worms and provides a localized and consistent gradient for them to follow. In this endpoint assay, trajectories are not recorded; instead, the number of worms reaching the target spot is documented every 5 min for up to 30 min post-stimulation to evaluate their responses to the introduced stimulus. This scoring method focuses on the culmination of navigational behavior, capturing the effectiveness of the worms' ability to detect and respond to the pheromone gradient over time. To ensure the reliability and robustness of our data, we repeated each assay 20 times across three distinct days. PHD-inhibited worms were prepared via the "Inhibition of PHD Neurons by HisCl Method". We remark that this C.I. quantification is not a projection onto the 2D plane, since worms at the target will stay still afterwards, but those not reaching the target will move in all directions. For example, a worm at the same $(x,y)$ coordinate as the pheromone location but vertically below the stimulus droplet (so the $z$ coordinate is different from the droplet) will continue to move around and will not be counted when computing the C.I. To control for vertical bias, the control solution and pheromone are placed on the same z-plane, preventing preference for movement toward the surface or bottom. Therefore, the C.I. reliably identifies worms at the target location in 3D space.

We prepared plates containing 0.3% (w/v) gellan gum in M9 buffer to generate a soft hydrogel substrate in which worms could enter and locomote within the gel volume, rather than being confined to surface crawling. Importantly, this "penetrable" behavior is not an intrinsic property of gellan gum, but depends strongly on gel concentration and the resulting mechanical properties. Higher gellan gum concentrations commonly used to cast firm gels (e.g., ~2–3% w/v) produce substantially stiffer matrices and are not expected to permit volumetric penetration. Accordingly, our statement refers specifically to the 0.3% (w/v) gellan-in-M9 plates used here, which we selected as the lowest concentration that maintained plate integrity while supporting robust 3D locomotion[118]. Gellan/Gelrite has been adopted for chemotaxis arenas in high-throughput screening and behavioral assays[119,120]. Penetrable gels and 3D habitats are known to elicit burrowing and other naturalistic behaviors[121–123].

## Optogenetics perturbation speed analysis

The optogenetics perturbation assay was conducted using Chrimson, a red-shifted channelrhodopsin, following protocols outlined by the literature[81,124,125], with slight modifications. All-trans-retinal (Sigma-Aldrich) was incorporated at a concentration of 500 μM into 6-cm diameter plates containing 20 mL of chemoattraction assay solution. One-day-old adult animals were placed on these plates and subjected to a red LED stimulus using a WormLab Tracking system (MBF Biosciences). Prior to stimulation, allow 5 worms a 5-min acclimation period on the assay plate. Exclusion criteria: (1) any worm that physically contacted another worm during the assay; (2) any worm that reached the plate edge. Across 150 imaged worms per sex, ~120 trajectories passed quality control and were analyzed. The LED stimulus was administered in a pattern of 5 s baseline record before stimulation, and 10–60 s on, followed by 7.5–30 s off, at 100% intensity across 6–8 cycles. Implement a 30-s inter-stimulus interval for PHD activation.

This provides sufficient time for the worm to return to a stable forward movement state. Worm tracking was conducted using the same WormLab system.

## Dissecting micro-behaviors under optogenetic perturbation

Experiments used 8-cm NGM agar plates seeded with a thin OP50 layer to prevent starvation-induced behaviors. Males were transferred from ATR+ and ATR-culture plates to an assay plate. The plate was mounted on a motorized stage controlled by a joystick. Individual males were tracked and recorded with a 10 × 0.3NA Nikon Plan Fluor objective at 10 Hz. Illumination was provided by dim, red-filtered light using a HOYA 52 mm R(25 A) filter to minimize disturbance to the worms. For the stimulation, we used a Sola Light Engine (Lumencor) light source and a 607/36 nm BrightLine excitation filter. The microscope's field diaphragm was adjusted such that only a small area was illuminated when the excitation light was on. We used the SOLA-SE2 software to control the light source. Prior to stimulation, allow a single worm for a 5-min acclimation period on the assay plate. The male was tracked manually using the motorized stage, and the following stimulation regime was applied: 30 s without light, followed by twenty 2-s tail pulses (2-s intervals), another 30-s dark period, then 30 s of head stimulation, and a final 30-s interval without light.

After postprocessing, we divided the recording into 2-s segments. For each 2-s segment, we tabulated the behavioral motif displayed by the male or the first behavioral transition that occurred during that segment. The following behavioral motifs and transitions were included: forward, backward, transition from forward to backward, transition from backward to forward movement, high-frequency direction transitions (marked by frequent direction changes within the 2-s window), and self-exploration (where the worm attempts to probe its own body). This analysis provides a detailed account of the nuanced responses of male *C. elegans* to optogenetic perturbation, offering insights into the micro-behavioral dynamics elicited by targeted neural stimulation.

Our analysis employs two analyses: The four-motif analysis (Fig. S6) focuses on core locomotion states (forward, backward) and transitions (forward to backward, backward to forward). In the four-motif analysis, transitions (e.g., forward-to-backward, backward-to-forward) were classified based on the initial and final directions observed within the 2-s window. For example, a segment with rapid oscillations between forward and backward movement was categorized as "forward-to-backward" if the net directionality shifted from forward to backward, or vice versa.

The six-motif analysis (Fig. 4) expands categorization to include self-exploratory and high-frequency direction transitions, which capture other behaviors that cannot be easily classified as the "four-motif". The high-frequency direction transitions, motif specifically identifies intervals with more than 2 direction changes within a 2-s window, representing a distinct exploratory strategy that cannot be unambiguously assigned to single transitions (forward to backward or backward to forward).

## Sex pheromone-induced pan-neuronal calcium imaging

For sex pheromone extraction, we processed 100 virgin *fog-2* female *C. elegans* by isolating them in 1 ml of NGM buffer within an Eppendorf tube, following a protocol slightly modified from the previous study[4]. These females were washed five times to clear bacterial contamination, then incubated in 100 μl of NGM or M9 buffer for 6 h. A parallel extraction process utilized for *C. remanei* involved similar steps but with only 5 females.

In Fig. 1C, to evaluate PHD activation upon exposure to the volatile pheromone, we performed calcium imaging with a custom-built spinning disk confocal microscope setup[48,73]. Utilizing the ZM9627 strain, which marks all neuronal nuclei with GCaMP6s and mNeptune[73] and incorporates *him-5(e1490)* mutation to enrich for

males. We also crossed ZM9627 with the CB5414 *srd-1(eh1)* line. Preparation for imaging involved isolating L4 males 15–20 h in advance, placing them on 10% agar NGM buffer slides.

The Thin Layer Micro-Diffusion (TLMD) method was adapted from Wan et al.[4] (see Fig. S9A). Neuronal excitation was detected by measuring changes in fluorescence intensity using a disposable worm-mounting and a chemical-loading device called the thin-layer micro-diffusion (TLMD) unit. The TLMD unit consisted of a 4% agarose pad (0.2 mm thick, 1.3 mm radius) sandwiched between two autoclave tapes. Before mounting, well-fed transgenic worms were transferred to an empty worm plate for 5 min to remove residual bacteria. A 0.2 µL of 1 mM levamisole was then applied to the agarose pad to partially immobilize the worms, which were positioned uniformly with their heads facing outward. A clean 22 × 40 mm coverslip was gently placed over the TLMD unit. At the 30-s time point, 8 µL of the test chemical solution was loaded. Due to capillary action, the solution rapidly filled the space between the slide and coverslip. Chemical diffusion began immediately (within milliseconds) as the micro-gel pad became surrounded by the loaded solution. Approximately 5 µl of 0.2 µm latex beads (Polysciences, Inc.) were added to aid immobilization.

Imaging was conducted with a 40×0.95 NA objective, achieving 200 Hz to record 10 volumes per second, with each comprising 20 optical slices. Pheromone or control buffer introduction occurred 30 s after initiating imaging, continuing for an additional 3.5–9.5 min. Subsequent processing and registration allowed for tracking PHD neuron nuclei using semi-automated methods in MAMUT[126,127]. Intensities from GCaMP and mNeptune channels were extracted for analysis within defined ROIs, calculating neuronal activity ratios and smoothing data with Savitzky-Golay filters. The activity was quantified by normalizing GCaMP fluorescence against mNeptune. We used the ratio $R = F(GCaMP)/F(mNeptune)$, after applying Savitzky–Golay filtering to each channel (polynomial order 1 and length 13) Relative intensity values ($\Delta R/R_0$) were calculated as $(R - R_0)/R_0$, where raw fluorescence intensity ratio ($R$) was extracted from the region of interest (ROI) of the target neuron in the calcium imaging data, and $R_0$ is the baseline fluorescence, defined as the mean fluorescence during the quiescent period. Analyses were performed using custom code.

In Fig. 5, our study employed the same custom-built spinning disk confocal microscope setup mentioned above and a sparse labeling markers pan-neuronal calcium imaging strain, ADS1046, which features a dual expression of the calcium indicator GCaMP6s and the red fluorescent protein mNeptune in all neuronal nuclei, alongside the *lite-1(ce314)* mutation to eliminate undesired light-induced neural activity. This strain also has sparse labeling markers (P*ift-20*:BFP, P*eat-4*:cyOFP1, P*unc-17*:mScarlet, P*acr-5*:BFP) for neuron identification. Intracellular calcium dynamics were quantified by tracking the calcium indicator GCaMP6s signals, which enhance green fluorescence intensity within cells. Concurrently, mNeptune signals were captured to serve as a reference. The intensity ratio of GCaMP to mNeptune (GFP/RFP) at each time point was plotted. To aid neuron identification, after the recording session, we acquired additional volumes using the following combination of excitation lasers and emission filters: 30% 640 nm 617/73 nm (200 Hz), 40% 561 nm 578 nm (200 Hz), 35% 488 nm 578 nm (200 Hz), 15% 488 nm (200 Hz), 95% 405 nm 440 or 450 nm (50 Hz).

We fit the activity of AVA neurons by a linear combination of AWA and PHD neurons by minimizing the least-squared error (Fig. 5F).

### Neuron tracking

Each neuron was automatically tracked through all 3D volumes with targettrack. We first applied a strong low-pass filter to blur the whole image and thresholded it to find the general location of the head or tail. After centering each volume by the region found above, we cropped a rectangular ROI that covered the whole region for 99% of the recording. Then, we manually identified AWA, ASI, AVA, AVE, and PHD neurons in each recording and annotated an average of 5 frames per

recording. We used targettrack[128] to automatically segment and track each neuron in every volume. We trained the 3DCN of targettrack for 10 K gradient steps with a batch size of 3. The quality of the tracking was manually verified after tracking.

The xyz coordinates were used to extract pixel intensities around neuronal nuclei. These changes were then standardized against the baseline pre-stimulation background, resulting in the relative intensity values ($\Delta R/R_0$) being displayed. Data representation is in the form of mean ± standard error of the mean (s.e.m.), processed using Graphpad Prism 6 and Microsoft Excel for comprehensive analysis. Custom Matlab code was utilized for graphing, with the $\Delta R/R_0$ average depicted by a black line and the s.e.m. by a gray-shaded envelope. The duration of stimulation is visually encoded on the plots with a pink color bar.

### Pan-neuronal calcium imaging with optogenetic activation

This experiment used the same sparse labeling makers, the pan-neuronal calcium imaging strain mentioned previously. For optogenetic activation, we expressed Chrimson using an *srd-1P*::Gal4(sk)::VP64 driver crossed to 15×UAS::Chrimson effector in the pan-neuronal imaging background; animals were reared ± ATR. During imaging, we delivered defined light stimuli; only ATR+ animals under light showed Chrimson-dependent responses. For the experiments, males were prepared on 10% agar NGM buffer slides. Using a spinning disc confocal setup, we navigated the challenge of simultaneous neuronal imaging and optogenetic stimulation of Chrimson, which responds to both 488 and 578 nm light, by implementing a strategy to exploit the GCaMP signal delay post-stimulation, as outlined in Lu et al.[129]. This involved interlacing five 10-s imaging/stimulation intervals with 50-s breaks, applying 12 and 35% intensities for the 488 nm and 561 nm lasers, respectively, while acknowledging potential mechanical stimulation from objective movement as a confounding variable. During each 5-s imaging/activation period, the piezo-controlled objective was moving, whereas it was not moving between these periods. We therefore cannot rule out mechanical stimulation that coincided with light stimulation as a confounding factor. To aid neuron identification, after the recording session, we acquired additional volumes using the following combination of excitation lasers and emission filters: 30% 640 nm 617/73 nm (200 Hz), 40% 561 nm 578 nm (200 Hz), 35% 488 nm 578 nm (200 Hz), 15% 488 nm (200 Hz), 95% 405 nm 440 or 450 nm (50 Hz).

The same pre-processing and neuron activity tracking via Targettrack was performed as outlined in "Neuron Tracking". For each neuron, we created 3 ground truth points, 100 volumes apart. Activities of PHD neurons in ATR+ and ATR- males were extracted and averaged. We calculated the correlation between the mean PHD activity in ATR+ males and activity traces of other neurons in both ATR+ and ATR- males. A subset of 12 neurons was identified across all males, and we used Generalized Procrustes analysis[130] and thin plate splines[131] to register all male tail neurons to a common coordinate space. These landmark neurons were: PHDR, PHDL, PLMR, PCAR, PCAL, HOA, PVY, PDC, PVX, VD12, VA10, and VA11. A visual inspection identified two regions that contain neurons highly correlated with PHD (correlation cutoff >0.55). Here we refer to these regions as ROI1 and ROI2 (see Figs. 5 and S6). AWA, ASI, AVA, and AVE are identified based on sparse labeling makers, positioning, and cell body size. The relative intensity values ($\Delta R/R_0$) are plotted following the methodology outlined in "Sex Pheromone-Induced Pan-neuronal Calcium Imaging in *C. elegans*" with minor modifications. We normalized the $\Delta R/R_0$ values for each neuron in the ATR+ and ATR- groups to a range of 0–1. The dashed line indicated the starting point of each light stimulation.

### Microfluidic chip design and manufacturing

The design of our microfluidic devices is accomplished using SolidWorks software, tailored to the specific requirements of our

experiments. The fabrication process utilizes both photolithography and polydimethylsiloxane (PDMS) soft lithography, as detailed in the literature[50,132]. Initially, 3-inch silicon wafers were prepared by coating with a layer of SU-8 photoresist. This layer was subsequently patterned through photolithography, transforming the wafer into a mold for PDMS casting. For constructs intended for male worm imaging, the SU-8 2015 photoresist was spun at 1450 rpm to achieve a thickness of 24–25 μm. Following PDMS casting, the microfluidic devices were bonded to glass coverslips using a plasma cleaner, employing settings of 50 mW power for 30 s to ensure a robust bond. This fabrication procedure is designed to produce precise microfluidic devices, enabling the simultaneous and individual delivery of stimulus to the head and tail regions of *C. elegans* males, while also facilitating concurrent calcium imaging. This chip allows for accurate delivery of stimulus to both the head and tail regions separately, with 4 inlets dedicated to each region and an additional outlet for waste. An optimized worm trapper adapter from a previous study[133], secures males for imaging while ensuring both ends remain exposed to the stimulus channels. Designed stoppers within the tail inlet channels prevent the worms from escaping during injection, without disrupting fluid flow. This approach allows for the detailed study of neural responses to localized stimuli in a controlled environment.

## Sex pheromone preparation for microfluidics chip-based experiments

To obtain large quantities of crude sex pheromone for microfluidics chip-based experiments, we modified the standard extraction protocol to avoid the time-consuming isolation of L4-stage *fog-2* females. We used 5-day-old *daf-22* mutant *C. elegans* hermaphrodites, which lack ascaroside synthesis, to ensure the absence of confounding neuronal responses from this water-soluble sex pheromone stimulus. Bleach-synchronized *daf-22* mutant worms were washed five times with M9 buffer. L1 stage arrest was induced by rotating worms in M9 buffer for 12 h before seeding on culture plates (8-cm NGM with OP50, approximately 500 L1 worms/plate). Adult worms were collected after 3 days (at the onset of egg-laying). Adults were washed from culture plates with M9 buffer and separated from eggs using the precipitation method (repeated 5–7 times until eggs were absent). Separated adults were transferred to new OP50-seeded plates, and this washing process was repeated for 6 days to exhaust self-sperm. Sex pheromone extraction was performed as described in "sex pheromone-induced pan-neuronal calcium imaging" with the following modification: instead of 100 worms/mL, we used 50 μl of packed worm volume per 1 mL of M9 buffer (see Fig. S10A–C). Three batches of extracted pheromone were tested for quality using a chemoattraction assay and combined to create a 100 ml homogenous stimulus for microfluidics chip-based experiments.

## Microfluidics chip-based experiment calcium imaging and data analysis

Young adult *C. elegans* males (PS10181) were loaded into a custom microfluidic chip that allows selective exposure of head and tail to odor solutions of choice. Solution administration was directed by an array of 8 memory-shape alloy valves (Takasago Fluidics Inc., Japan), individually addressed by an ESP32 microcontroller. Imaging was performed using a Leica DM6000 B microscope equipped with a X-Cite XLED1 Multi-Triggering LED illumination system (Excelitas Technologies) and iXon EMCCD detectors (Oxford Instruments Andor). Image acquisition and microfluidic stimulation administration were controlled and synchronized by a custom Python graphic interface using the Pycro-Manager interface for the Micro-Manager microscope control software[134]. Time-

lapse image segmentation was performed using the pixel classification function of the Ilastik software[135], and intensity values of GCaMP and RFP were extracted using a custom Jupyter notebook. The calcium imaging traces were calculated as the ratio of GCaMP6s/RFP fluorescence to correct for out-of-plane imaging artifacts. Because of the high background activity of the AVA neuron, baseline values for the calculation of the $\Delta R/R_0$ metric were estimated using the peakutils Python package. The relative intensity values ($\Delta R/R_0$) are plotted following the methodology outlined in "Sex Pheromone-Induced Pan-neuronal calcium imaging in *C. elegans*," with minor modifications. To quantify the effect of suppression/excitation on the AVA neuron, we performed area-under-curve (AUC) analysis. We first compute the baseline AVA activity by averaging over 3 min before stimulation given at the most dilute concentration, and subtract this baseline value from all signals. Then we integrate the area under the curve for each worm data for 2 min after the stimulation. For suppression, we only account for time intervals when the signal is lower than the baseline value; for excitation, we only account for time intervals when the signal is higher than the baseline value, so that to naturally terminate the integration of area when the signal disappears. In Fig. 6D, we compute the linear combination of AVA neuron activities with only head or tail stimulation ($\text{AVA}_{head}+\text{AVA}_{tail}$) and compare that with the AVA neuron activity with simultaneous head and tail stimulation ($\text{AVA}_{dual}$).

## Concentration field simulation

The concentration evolution inside the petri dishes was simulated using the FEM. The simulated plates consist of a top layer of 4 mm thick air and a bottom layer of 9 mm thick agar. The diffusion coefficient of the sex pheromone was estimated $1.6 mm^2/s$ in the air and $10^{-3} mm^2/s$ in the agar. The initial condition was set as a concentration at the surface of the droplet $c_0 = 0.01 mol/m^3$ while $c_0 = 0$ elsewhere. No-flux boundary condition was applied to the walls of the plate. For the dynamical simulations of worm searching, we take the 2D cross-section of the concentration field at $z = -0.5$ mm inside the agar (the air-agar interface is at $z = 0$).

## Navigation simulation

The navigation processes were simulated using an in-house extension of the HOOMD-Blue package. Mathematically, worm locomotion dynamics are described by the following equations:

$$d\boldsymbol{r} = U\boldsymbol{q}dt, \boldsymbol{q} = (\cos\theta, \sin\theta), d\theta = \sqrt{2D_R}dW_S + \gamma dW_T \tag{3}$$

Here $\boldsymbol{r}$ is the position of the worm, U the speed magnitude, **q** the orientation direction, $D_R$ the rotary diffusivity of the orientation angle $\theta$. $dW_S$ and $dW_T$ denote the standard Brownian motion on a circle $-\pi < \theta < \pi$ and a Poisson jump process, respectively. $\boldsymbol{\gamma}$ is the intensity of the jump process, i.e., the tumbling rate. It has been observed[136] that during tumbles (large angle turns), the turning angle distribution displays a peak near π. Empirically, we represent the turning angle distribution as a mixture of a uniform distribution and a wrapped Gaussian distribution:

$$T(\theta, \theta') = \alpha T_0 + (1-\alpha)T_1 \tag{4}$$

Here the turning kernel $T(\theta, \theta')$ is the probability of turning from $\theta'$ direction to $\theta$ direction. $T_0 = 1/(2\pi)$ is the uniform distribution and $T_1 = e^{-\frac{(\theta-\pi)^2}{2\sigma^2}}$ is a wrapped Gaussian distribution.

We adopt the following functional forms to reflect the above observed sensory neuron dynamics:

$$f_H = \frac{\dot{C}}{\dot{C}_0} \frac{1 + tanh(log(C/C_0))}{2},$$

$$f_T = \begin{cases} e^{-(\log(C/C_0))^2} & \text{if } C \leq C_0 \\ 1 - (\log(C/C_0))^2 & \text{if } C > C_0 \end{cases} \tag{5}$$

In the simulations, we fix $D_R = 0.02$, $\eta = 1$, $k_1 = 1/60$, $k_2^{H,T} = 1$.

### Reporting summary

Further information on research design is available in the Nature Portfolio Reporting Summary linked to this article.

### Data availability

The data generated in this study have been deposited at the GitHub repository (pheromone-traj-analysis) https://github.com/edmondztt/pheromone-traj-analysis[137]. Calcium imaging and behavioral datasets supporting Figs. 1, 4, 7 and Supplementary Figs. S1, S3, S5, S11 are available from the CaltechDATA repository https://doi.org/10.22002/dp6gb-ky687[138]. Source data are provided with this paper.

### Code availability

The code used for data processing and analysis in this study is publicly available at https://github.com/edmondztt/pheromone-traj-analysis[137].

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

## Acknowledgements

We are grateful to David Aguirre, Wilber Palma, Stephanie Nava, and Zachary Blumenfeld for their assistance during the early stages of this project. We acknowledge Natalie Sim and Elena Ruiz for data analysis. We also thank Helena Casademunt and Nicholas Markarian for their discussion and suggestions during the manuscript preparation. This work was supported by: R01 NS113119 (P.W.S., A.D.T.S.), Tianqiao and Chrissy Chen Institute for Neuroscience senior postdoc fellowship (X.W.), Tianqiao and Chrissy Chen Institute for Neuroscience postdoc innovator grant (X.W.), and Tianqiao and Chrissy Chen Institute for Neuroscience Systems Neuroscience Awards (P.W.S.). P.W.S. is Bren Professor of Biology. The *Caenorhabditis* Genetics Center provided some strains.

## Author contributions

X.W. and P.W.S. conceived the study and designed the experiments. X.W. and V.S. performed the behavioral assays, imaging, and optogenetics experiments. T.Z. and X.W. designed the computational model, performed the simulation, and data analysis. X.W. and A.G. performed the microfluidics chip-based experiment. C.F.P. provided a whole-brain imaging data analysis platform. X.W., T.Z., and P.W.S. wrote the initial draft of the manuscript. All authors discussed the results and contributed to the final version of the manuscript. P.W.S., A.D.T.S., and J.F.B. supervised the project.

## Competing interests

The authors declare no competing interests.
