## [Transparent Peer Review file · Nature Communications]

Efficient pheromone navigation via antagonistic detectors in *Caenorhabditis elegans* male

Corresponding Author: Professor Paul Sternberg

Version 0:

Reviewer comments:

Reviewer #2

(Remarks to the Author)

The authors have used a delightful variety of methods to dissect the role of anterior and posterior chemosensory neurons in the ability of *C. elegans* males to detect and respond to sex pheromone. The findings are noteworthy and interesting; they have the potential to change how we think about navigation toward pheromones. The manuscript has been revised and edited and figures have been improved and reordered. These revisions address some but not all of the concerns raised by myself and the other reviewers. Nonetheless, the text and the presentation of the data lack sufficient clarity and consistency to recommend publication. It is beyond my capacity as a reviewer to identify all of the elements requiring further attention. To aid the authors, however, I identify some themes that hamper readability and undermine confidence in the rigor of the study.

The text is dense and meandering, in places For instance, the manuscript primarily concerns the response of *C. elegans* males to sex pheromone and the role of the anterior AWA and the posterior and male-specific PHD sensory neurons in this behavior. However, the reader is not informed of this focus for the study or that PHD is a male-specific sensory neurons in the Abstract or Introduction. Another example is the treatment of sexually dimorphic expression of SRD-1, which is emphasized in the text surrounding Figure 1. Imagine if the multiple lines of evidence that SRD-1 is expressed in PHD were consolidated into a single figure (currently in Figure 1 and Figure 2A). This could precede a clear statement in the text that the rest of the manuscript will focus on male *C. elegans*, using SRD-1 as a tool to dissect behavioral and cellular responses to sex pheromone.

The text and figure legends rely on shorthand to refer to strains and reagents, compromising clarity and rigor. An example is the phrase "Using the odr-10p cGAL driver to activate AWA neurons" (line 218). This is not accurate, as written. The cGAL driver cannot by itself activate anything without an effector. From context, I am guessing that the authors mean that they combined an odr-10p cGAL driver with a UAS Chrimson effector and then used light to activate AWA neurons. This is not the only instance in which the authors are using shorthand to describe their experiments, leaving the reader to guess what reagents were used for each experiment and the value of this experimental design.

The main figures are improved, but remain dense and imprecise. Many figures and figure legends have at least one error, omission, or design element that compromises clarity. Below are specific examples of aspects of the figures and legends that impair the reader's ability to evaluate the evidence.

- Figure 1A: The image shows green fluorescence, but the legend only mentions mCherry and RFP expression
- Figure 1C: The legend says that shaded areas indicate the s.e.m. of calcium traces. No shaded areas are visible in the panel on the far left and the one of the far right. In the far right, the only visible trace is rendered in black while in the other panels the traces are rendered in green.
- Figure 2: Tic labels are too small to read
- Figure 3: What is meant by "total velocity"?
- Figure 3C: The pink/red bars are not mentioned in the figure legend and what the authors mean by 30 samples is not clear.
- Figure 3Dd: typo/missing words "represent the std."
- Figure 4: Tic labels are too small to read
- Figure 4Cd: "Fourier analysis of $V(t)$ from Figure 2A and B" Figure 2A and 2B do not contain $V(t)$ traces.

Missing control experiments, unsubstantiated claims

- Along with optogenetic tools for activating neurons, the authors also use chemogenetic silencing by application of histamine to animals expressing HisCl in specific neurons. Whereas animals expressing HisCl are tested in the presence and absence of histamine to control for non-specific effects of HisCl expression, the effect of histamine itself on pheromone responses is not tested. Without this control, the authors cannot exclude the possibility that histamine itself, rather than histamine-induced silencing, is responsible for the phenotype observed.
- What is the evidence that gellan gum “better mimics natural conditions”? (line 218). It seems that the authors embedded animals in a gellan gum layer (Methods), the authors should either show that animals moved vertically in this environment or cite prior studies demonstrating this.

(Remarks on code availability)

Not applicable

Reviewer #3

(Remarks to the Author)

This paper reports a detailed analysis of the neural mechanism underlying a *C. elegans* male's ability to follow a volatile female-produced pheromone to find the female mating target. The investigators find that the mechanism involves integration of the outputs of two different sensory neuron classes that both express the same receptor for the pheromone but have different response properties to it. The investigative approach exploits all of the experimental and other advantages provided by *C. elegans* to tackle the general problem, widespread among animals, of how to use the information in a rapidly changing and likely shallow gradient of a volatile substance to find the source of the gradient. The methods include the powerful tools available to measure, activate, or inhibit single neuron activity, determine gene expression patterns, behavioral assays including behavioral tracking of individual worms, and known connectivity. A detailed analysis at this depth could not be carried out at present in another organism. The authors provide support for the sufficiency of their interpretation of the mechanism by mathematical modeling. The conclusions are interesting and convincing.

(Remarks on code availability)

Reviewer #4

(Remarks to the Author)

This manuscript by Wan et al. has been extensively reviewed by three reviewers, who provided many valuable critiques and suggestions. As a new reviewer in this second round, I did not have the opportunity to read the previous version, but judging from the rebuttal, it is clear that the authors have made substantial revisions to address all points raised. In my opinion, the three major concerns from Reviewer 1 have been sufficiently ameliorated. The decision to remove the section related to orthokinesis, which appeared to be the most problematic, is also appropriate.

In its current form, I do not find any major issues that would require further substantial revision. Nevertheless, I have two minor points that the authors may wish to consider discussing:

1. Variability and heterogeneous SRD-1 expression:

The authors attributed much of the variability observed in the study (e.g., Lines 242, 274, and 320) to the partial penetrance of SRD-1 expression in only ~40% of PHD neurons, as reported by the Murphy group. It is not clear to me whether the positive expression of a transgene (such as *Chrimson*) driven by the *srd-1* promoter in the various strains used in this study can serve as an indicator of SRD-1 expression. If so, the lack of SRD-1 expression in PHD neurons should not be invoked to explain the variability (I would assume that only worms with positive marker expression in PHD neurons were selected for experiments). If not, is there a way to determine SRD-1 expression directly—for instance, by fluorescently tagging the SRD-1 protein and comparing PHD neuron function with or without SRD-1? This approach would be useful for specifically testing SRD-1 function in PHD neurons in future studies. In addition, if SRD-1's function in PHD is important for mate-finding behavior, it would be interesting to discuss how such heterogeneous expression could be maintained evolutionarily.

2. Spatial separation of AWA and PHD neurons on chemotaxis:

I am not entirely convinced that the spatial separation of these two neurons—in the head and tail, respectively—has no impact on chemotaxis, as the authors argue but this has not been tested. When a worm crawls toward a pheromone source in a gradient, the head neuron AWA would experience a higher pheromone concentration than the tail neuron PHD, and vice versa when crawling away. According to this paper, where AWA and PHD neurons have distinct concentration thresholds, I can't help wondering if the following could be true: when the worm moves in the correct direction, AWA experiences stronger stimulation than PHD, AWA is suppressed, and the worm continues moving forward; conversely, when the worm moves in the wrong direction, AWA receives weaker stimulation than PHD, AWA is activated, and the worm reorients. Although the authors argue that the distance between the head and tail (~1 mm) is too small to matter, this likely depends on how steep the gradient is—particularly when the worm is near the female, where the gradient may be sharp enough to influence the “final sprint.” It would be interesting to know whether the mathematical model could test this hypothesis, or whether it could be experimentally examined by exposing AWA and PHD to different pheromone concentrations using the microfluidic setup shown in Figure 6A and monitoring AWA activity.

Both points are merely suggestions and should not be taken as conditions for publication. The authors have undergone a lengthy review process and have addressed all previous reviewers' concerns. It would not be fair to request additional revision from a new reviewer at this stage to cause further delay. If Reviewers 2 and 3 are satisfied with the revisions, I would recommend expedited publication, as this paper provides a significant contribution and will be of broad interest to the field.

(Remarks on code availability)

Version 1:

Reviewer comments:

Reviewer #2

(Remarks to the Author)

The authors have improved the presentation of the findings in the text, figures, and figure legends. There are some concerns that remain regarding methodology and these concerns bear on the rigor of the study and its presentation:

1) Gellan Gum — The authors claim that this material by itself creates a "penetrable medium" and, by inference that Agar does not. The authors have added text and citations in support of this claim, but simply citing these other studies is not convincing. Why? Because it is not only the material itself, but also the concentration of the material. Fryer et al use 2.5% (w/v) gellan gum and do not report that this medium is penetrable. By contrast, Wan and Sternberg use 0.3% (w/v) gellan gum to generate a droplet in which single worms can thrive. To avoid misleading others in the field about the performance of gellan gum as an alternative growth medium, please revise the methods to be more precise.

2) Throughout the manuscript the authors use "velocity" to describe the rate of animal movement. Strictly speaking, velocity is a vector quantity consisting of both direction and rate. Please consider revising the language to use the term "speed" in its place.

(Remarks on code availability)

Reviewer #4

(Remarks to the Author)

The revised manuscript has addressed all the points raised by me and the other reviewers, and it is much improved in both clarity and rationale. I especially appreciate the addition of Fig. S3C, which quantitatively addresses my previous concern. Only one very minor point: the "red line" in Fig. S3C does not appear red, at least in the version of the PDF I received.

(Remarks on code availability)

REVIEWER COMMENTS

Reviewer #2 (Remarks to the Author):

The authors have used a delightful variety of methods to dissect the role of anterior and posterior chemosensory neurons in the ability of *C. elegans* males to detect and respond to sex pheromone. The findings are noteworthy and interesting; they have the potential to change how we think about navigation toward pheromones. The manuscript has been revised and edited and figures have been improved and reordered. These revisions address some but not all of the concerns raised by myself and the other reviewers. Nonetheless, the text and the presentation of the data lack sufficient clarity and consistency to recommend publication. It is beyond my capacity as a reviewer to identify all of the elements requiring further attention. To aid the authors, however, I identify some themes that hamper readability and undermine confidence in the rigor of the study.

The text is dense and meandering, in places For instance, the manuscript primarily concerns the response of *C. elegans* males to sex pheromone and the role of the anterior AWA and the posterior and male-specific PHD sensory neurons in this behavior. However, the reader is not informed of this focus for the study or that PHD is a male-specific sensory neurons in the Abstract or Introduction. Another example is the treatment of sexually dimorphic expression of SRD-1, which is emphasized in the text surrounding Figure 1. Imagine if the multiple lines of evidence that SRD-1 is expressed in PHD were consolidated into a single figure (currently in Figure 1 and Figure 2A). This could precede a clear statement in the text that the rest of the manuscript will focus on male *C. elegans*, using SRD-1 as a tool to dissect behavioral and cellular responses to sex pheromone.

Reply: We have revised the Abstract and Introduction to explicitly state that AWA is a sex-shared head sensory neuron and PHD is a male-specific tail sensory neuron.

Page 1: “*Contrary to this model of simple spatial comparison, C. elegans males employs an antagonistic strategy, comparing inputs from sex-shared head (AWA) and male-specific tail (PHD) sensory neurons with distinct response properties.*”

Page 3: “*In this study, we discover that the male tail-specific neuron, PHD, in addition to AWA, expresses SRD-1 and responds to the sex pheromone as well.*”

In the manuscript, we consolidate the evidence for sexual dimorphic *srd-1* expression in a single figure (Fig. 1). In Fig. 2A, we validate the split cGAL–UAS driver (*lin-48p::NLS::cGAL[DBD]; srd-1p::NLS::cGAL[AD]*) by crossing it to $15\times$ UAS::GFP, which drives GFP in PHD in male tails. We placed this panel in Fig. 2 because its primary purpose is tool validation and experimental control for the manipulation assays (“driver → effector → behavior”), not expression mapping. While the result corroborates *srd-1* expression in PHD (since *lin-48* is known to be expressed in PHD neurons), its central role is to establish a PHD-specific split-cGAL driver used immediately in Fig. 2 for functional inhibition/activation; therefore, it belongs with the manipulation experiments rather than the expression summary in Fig. 1.

The text and figure legends rely on shorthand to refer to strains and reagents, compromising clarity and rigor. An example is the phrase “Using the odr-10p cGAL driver to activate AWA neurons” (line 218). This is not accurate, as written. The cGAL driver cannot by itself activate anything without an effector. From context, I am guessing that the authors mean that they combined an odr-10p cGAL driver with a UAS Chrimson effector and then used light to activate AWA neurons. This is not the only instance in which the authors are using

shorthand to describe their experiments, leaving the reader to guess what reagents were used for each experiment and the value of this experimental design.

Reply: We have revised the text and figure legends to explicitly state the driver, UAS effector, and stimulation/ATR conditions throughout.

Page 5: *“Using an odr-10p cGAL driver crossed to UAS::Chrimson, optogenetic activation of AWA in both sexes produced no detectable behavioral change, indicating that AWA activation alone does not drive forward locomotion or that odr-10p cGAL driver yielded insufficient Chrimson expression for activation (Fig. S4B).”*

Beyond this instance, all occurrences of shorthand have been corrected to name (i) the specific cGAL driver(s), (ii) the UAS effector(s), and (iii) illumination and ATR or HisCl status in both text and figure legends, removing ambiguity about reagents and experimental design (highlight with red color in the file):

Page 4: *“PHD-inhibited males refers to males expressing HisCl in PHD via split-cGAL (lin-48dP::NLS::cGAL[DBD]; srd-1P::NLS::cGAL[AD]) crossed to 15×UAS::HisCl::SL2::GFP and exposed to 10 mM histamine (See Method).”*

Page 6: *“Under matched light stimulation, ATR– males exhibited no effect.”*

Method, Page 17: *“ For PHD-specific inhibition we used the split-cGAL driver (lin-48dP; srd-1P) crossed to 15×UAS::HisCl::SL2::GFP; animals on 10 mM histamine (NGM-HA) are referred to as ‘PHD-inhibited,’ whereas histamine-free condition worms served as controls”*

Method, Page 21: *“For optogenetic activation we expressed Chrimson using an srd-1P::Gal4(sk)::VP64 driver crossed to 15×UAS::Chrimson effector in the pan-neuronal imaging background; animals were reared ± ATR. During imaging we delivered defined light stimuli; only ATR+ animals under light showed Chrimson-dependent responses.”*

Figure 2 figure legend, Page 26: *“ (A) Split cGAL construction of a PHD neurons-specific driver and validation of SRD-1 expression. Male tail fluorescence from lin-48dP::NLS::cGAL(DBD); srd-1P::NLS::cGAL(AD) crossed to 15×UAS::GFP demonstrates GFP labeling of PHD, verifying SRD-1 expression and providing a driver for targeted PHD manipulation.”*

Figure 3 figure legend, Page 28: *“ (A)(B) Schematic of optogenetics with region-specific genetic targeting: (A) head and tail region AWA, ASI and PHD neurons (srd-1P–cGAL > UAS::Chrimson) versus (B) tail PHD neurons (split cGAL: lin-48dP::NLS::cGAL[DBD]; srd-1P::NLS::cGAL[AD] > UAS::Chrimson). Light stimulation in freely moving males tests locomotor consequences of activating specific neurons.”; “(B) PHD neurons were activated by expressing Chrimson via a split-cGAL driver (lin-48dP::NLS::cGAL[DBD]; srd-1P::NLS::cGAL[AD]) crossed to 15×UAS::Chrimson; animals were treated with or without ATR and stimulated with light during the assay. ATR– and light-off served as negative controls.”*

Figure 4 figure legend, Page 29: *“ (A) Summary of all characterized micro-behavioral and their portion among all behaviors. With srd-1 drivers and Chrimson effectors in cGAL-UAS lines (srd-1P–cGAL > UAS::Chrimson) combined with region-specific light sources, we precisely targeted the head (AWAs and ASIs) or tail regions (PHDs) of freely moving males to analyze locomotion effects and characterize specific micro-behaviors. Behavioral segments were classified in 2-s windows as forward (fwd), backward (bwd), fwd→bwd reversal, bwd→fwd reversal, high-frequency direction transitions, or self-exploratory. High-frequency direction transitions, defined as >2 direction changes within a 2-s window, reflecting an exploratory strategy that cannot be unambiguously assigned to fwd→bwd or bwd→fwd. Self-exploratory, in which the worm engages in near-body probing behavior. Short labels used in panels: bwd→fwd reversal, fwd, fwd→bwd reversal, bwd, high-freq transitions, self-exploratory. (B) Tail-restricted illumination increases high-frequency transitions during light stimulation. All animals carried srd-1P–cGAL > 15×UAS::Chrimson and were reared ± all-trans-retinal (ATR).*

*To probe tail-region circuits (PHDs), we delivered posterior-restricted light (illumination geometry and parameters in Methods). During light-on, ATR+ animals exhibited a robust increase in high-frequency direction transitions; this effect was absent in ATR- controls and in light-off trials. Behavior returned toward baseline after light-off. (C) Head-restricted illumination suppresses transitions during light stimulation and promotes transitions and self-exploration after light-off. Using the same *srd-1P-cGAL > UAS::Chrimson* genotype, we applied head-restricted light to engage head-region circuits (including AWAs/ASIs). In ATR+ animals, light-on suppressed direction transitions (both *bwd*→*fwd* and *fwd*→*bwd*). Following light-off, both transitions rebounded above baseline, and animals displayed increased self-exploratory “body-probing” behavior (tail pressed against/near the body). No significant changes were observed in ATR- or light-off controls.”*

Figure 5 figure legend, Page 32: “(B)-(C) For optogenetic activation and calcium imaging we expressed *Chrimson* by crossing an *srd-1P::Gal4(sk)::VP64* driver to a $15\times UAS::Chrimson$ effector in the pan-neuronal imaging background. Animals were reared \pm ATR, and defined light stimuli were delivered during imaging; *Chrimson*-dependent responses were observed only in ATR+ light-on trials.”; “(D)-(F) Pheromone odor stimulation and calcium imaging. All panels use the pan-neuronal imaging strain and are exposed to volatile sex-pheromone extract; stimulus and the delivery method/protocol are detailed in Methods.”

Figure S2 figure legend, Page 40: “(B) Worm counts at each spot across time points. Different navigational deficits in PHD-inhibited males across different exploration tasks: in weak signal tasks, their even distribution between test and control spots led to a low chemotaxis index; in 3D tasks, the low index resulted from fewer males reaching the target. Inhibition was achieved by crossing the indicated PHD neurons-specific driver line to $15\times UAS::HisCl::SL2::GFP$ and assaying on 10 mM histamine; *him-5* and genotype-matched animals maintained on histamine-free plates served as controls (See Methods).”

The main figures are improved, but remain dense and imprecise. Many figures and figure legends have at least one error, omission, or design element that compromises clarity. Below are specific examples of aspects of the figures and legends that impair the reader's ability to evaluate the evidence.

- Figure 1A: The image shows green fluorescence, but the legend only mentions mCherry and RFP expression

Reply: Thank you for pointing out the error. The panel displays GFP, not RFP. We corrected the legend to match the image (See Method, Key Resource Table: syEx1972 [*srd-1P::mCherry*]; syEx1974 [*unc-119::gfp*]).

- Figure 1C: The legend says that shaded areas indicate the s.e.m. of calcium traces. No shaded areas are visible in the panel on the far left and the one of the far right. In the far right, the only visible trace is rendered in black while in the other panels the traces are rendered in green.

Reply: All subpanels were plotted as mean \pm s.e.m.; however, in the far-left and far-right panels the PHD activity was essentially absent, so the traces lie near zero and the s.e.m. envelope is extremely tight, making the shading difficult to see. To avoid confusion, we added a zoomed-in view of the far-left and far-right conditions in Fig. S1 to make the (very small) s.e.m. shading visible.

In the previous version, WT traces were green and mutant traces were black, which may have been confusing. We have re-plotted the figure using green for all calcium traces.

Figure S1D, Page 39: “(D) Zoomed view of Figure 1C first and last panel (*y*-axis -5 to 5). Pan-neuronal calcium imaging of male PHD neurons expressing *rgef-1P::GCaMP6s*, with *rgef-1P::mNeptune* marking all neuronal somata. WT males received M9 buffer (control); *srd-1* mutant males received volatile sex pheromone. Traces show mean $\Delta R/R_0$ ($R = \text{GCaMP}/\text{mNeptune}$ fluorescence ratio) \pm s.e.m.; $n = 5-7$ animals per condition. The green trace denotes the mean calcium ratio ($\Delta R/R_0$). Shading denotes pre-stimulus (white), M9 control buffer application (yellow), and pheromone application (pink).”

- Figure 2: Tic labels are too small to read

Reply: Increased axis tick label and title fonts across Fig. 2 to ensure legibility at print scale.

- Figure 3: What is meant by "total velocity"?

Reply: It just means the velocity of the worm. In the context here, we decomposed the velocity into radial and tangential components for analysis. We change the wording now to “velocity” since the discussions on radial velocity analysis has been removed for clarity and focus of this manuscript.

- Figure 3C: The pink/red bars are not mentioned in the figure legend and what the authors mean by 30 samples is not clear.

Reply: We fixed the legend to define the color coding and replaced “30 samples” with the exact worm count and unit of replication. We also clarified exclusions in Methods.

1. **Shaded pink/red bars show the stimulus-on periods.**
2. **Sample size clarified:** ~120 worms per sex (30 assay × 5 worms per assay)
3. **Exclusions added:** worms that contacted another worm or reached the plate edge were excluded.

Figure 3 figure legend, Page 28: “(c) $V(t)$ averaged over about 120 worms.” ; “Pink/red bars indicate periods of light stimulation.”

Methods, Page 19: “Prior to stimulation, allow 5 worms a 5-minute acclimation period on the assay plate. Exclusion criteria: (1) any worm that physically contacted another worm during the assay; (2) any worm that reached the plate edge. Across 150 imaged worms per sex, ~120 trajectories passed quality control and were analyzed.”

- Figure 3Dd: typo/missing words “represent the std.”

Reply: We corrected the wording and standardized error display to “(c), (d) Shaded regions indicate \pm s.d.; solid lines are means.”

- Figure 4: Tic labels are too small to read

Reply: Increased axis tick and title fonts across Fig. 4 for legibility.

- Figure 4Cd: “Fourier analysis of $V(t)$ from Figure 2A and B” Figure 2A and 2B do not contain $V(t)$ traces.

Reply: Corrected the reference to the figure that actually contains $V(t)$.

Figure 3 figure legend, Page 27: “Fourier analysis of $V(t)$ from Fig. 3A–B.”

Missing control experiments, unsubstantiated claims

- Along with optogenetic tools for activating neurons, the authors also use chemogenetic silencing by application of histamine to animals expressing HisCl in specific neurons. Whereas animals expressing HisCl are tested in the presence and absence of histamine to control for non-specific effects of HisCl expression, the effect of histamine itself on pheromone responses is not tested. Without this control, the authors cannot exclude the

possibility that histamine itself, rather than histamine-induced silencing, is responsible for the phenotype observed.

Reply: Added this control to Figure 2, Page 25. *him-5* males with same concentration histamine treatment (10 mM). No histamine specific effect on the chemotaxis assays.

Figure 2, Page 26 : “Males expressing *lin-48dP::NLS::cGAL (DBD)*; *srd-1p::NLS::cGAL (AD) (driver)* × *UAS::HisCl (effector)* were inhibited by *HisCl* to silence PHD. Controls were the same genotype on without *HisCl*, and WT *him-5* with or without *HisCl*, as indicated. (C)-(E) Baseline chemotaxis: PHD-inhibited males locate sex pheromone comparably to controls. (E) Sensitivity test (10× diluted pheromone): PHD inhibition significantly reduces detection and targeting performance. (G) 3D navigation (gellan gum/gel arena): PHD-inhibited males show impaired source finding relative to controls.”

Page 4: “Thus, observed effects reflect histamine-induced PHD silencing, rather than genetic background or histamine alone.”

• What is the evidence that gellan gum “better mimics natural conditions”? (line 218). It seems that the authors embedded animals in a gellan gum layer (Methods), the authors should either show that animals moved vertically in this environment or cite prior studies demonstrating this.

Reply: We agree that this claim should be supported. Semi-fluid gellan gum media have been shown to permit three-dimensional distribution and free movement of nematodes and food bacteria, thus more closely approximating soft, structured natural substrates than hard agar surfaces. Accordingly, we now cite Brinke et al., 2011 and note that gellan/Gelrite is also used for chemotaxis arenas in recent work (Fryer et al., 2024). Related 3D gel/habitat studies (Lesanpezeshki et al., 2019; Beron et al., 2015; Guisnet et al., 2021; Wan et al., 2025) similarly show that penetrable, 3D environments reveal naturalistic behaviors not observed on flat agar.

Methods, Page 19: “We used gellan gum to provide a semi-fluid, penetrable matrix that supports 3D distribution and locomotion, which is closer to the worms’ native substrates than regular agar [Brinke 2011]; gellan/Gelrite has been adopted for chemotaxis arenas in high-throughput screening and behavioral assays [Fryer 2024; Wan 2025]. Penetrable gels and 3D habitats are known to elicit burrowing and other naturalistic behaviors [Lesanpezeshki 2019; Beron 2015; Guisnet 2021].”

1. Brinke, M., Heininger, P., & Traunspurger, W. (2011). A semi-fluid gellan gum medium improves nematode toxicity testing. *Ecotoxicology and Environmental Safety*, 74(7), 1824–1831. <https://doi.org/10.1016/j.ecoenv.2011.07.007>
2. Fryer, E., et al. (2024). A high-throughput behavioral screening platform for measuring chemotaxis by *C. elegans*. *PLOS Biology*, 22(6): e3002672. <https://doi.org/10.1371/journal.pbio.3002672>
3. Lesanpezeshki, L., et al. (2019). Pluronic gel-based burrowing assay for rapid assessment of neuromuscular health in *C. elegans*. *Scientific Reports*, 9:15246. <https://doi.org/10.1038/s41598-019-51608-9>
4. Beron, C., Vidal-Gadea, A. G., Cohn, J., Parikh, A., Hwang, G., Pierce-Shimomura, J. T. (2015). The burrowing behavior of the nematode *Caenorhabditis elegans*: A new assay for the study of neuromuscular disorders. *Genes, Brain and Behavior*, 14(4), 357–368. <https://doi.org/10.1111/gbb.12217>
5. Guisnet, A., et al. (2021). A three-dimensional habitat for *C. elegans* environmental enrichment. *PLOS ONE*, 16(1): e0245139. <https://doi.org/10.1371/journal.pone.0245139>
6. Wan, X., Sternberg, P. W., et al. (2025). GelDrop array high-throughput screening in nematodes. *microPublication Biology*. <https://doi.org/10.17912/micropub.biology.001809>

Reviewer #2 (Remarks on code availability):

Not applicable

Reviewer #3 (Remarks to the Author):

This paper reports a detailed analysis of the neural mechanism underlying a *C. elegans* male's ability to follow a volatile female-produced pheromone to find the female mating target. The investigators find that the mechanism involves integration of the outputs of two different sensory neuron classes that both express the same receptor for the pheromone but have different response properties to it. The investigative approach exploits all of the experimental and other advantages provided by *C. elegans* to tackle the general problem, widespread among animals, of how to use the information in a rapidly changing and likely shallow gradient of a volatile substance to find the source of the gradient. The methods include the powerful tools available to measure, activate, or inhibit single neuron activity, determine gene expression patterns, behavioral assays including behavioral tracking of individual worms, and known connectivity. A detailed analysis at this depth could not be carried out at present in another organism. The authors provide support for the sufficiency of their interpretation of the mechanism by mathematical modeling. The conclusions are interesting and convincing.

Reply: We thank the reviewer for the thoughtful and generous assessment of our work and its significance for understanding navigation in shallow, rapidly fluctuating odor gradients. In keeping with the reviewer's emphasis on clarity and rigor, our revision further strengthens the manuscript in several ways:

Clarity of experimental design. We rewrote all relevant captions and Methods passages to explicitly state the driver \times effector genotype, ATR status, and light parameters for optogenetic experiments, and to define the HisCl + histamine condition at first use for inhibitory manipulations.

Figure corrections and legend precision. We corrected the annotation issues and the font size suggestions.

Added a WT animal control for histamine. We added a control experiment showing that 10 mM histamine alone does not affect pheromone chemotaxis (Fig. 2), ensuring that phenotypes in HisCl experiments reflect neuron silencing rather than histamine per se.

Report the time-dependent concentration difference at 1 mm predicted by the two-layer concentration simulation (mathematical model).

Reviewer #4 (Remarks to the Author):

This manuscript by Wan et al. has been extensively reviewed by three reviewers, who provided many valuable critiques and suggestions. As a new reviewer in this second round, I did not have the opportunity to read the previous version, but judging from the rebuttal, it is clear that the authors have made substantial revisions to address all points raised. In my opinion, the three major concerns from Reviewer 1 have been sufficiently ameliorated. The decision to remove the section related to orthokinesis, which appeared to be the most problematic, is also appropriate.

In its current form, I do not find any major issues that would require further substantial revision. Nevertheless, I have two minor points that the authors may wish to consider discussing:

Reply: We thank Reviewer #4 for carefully evaluating our revised manuscript and for the supportive assessment. We appreciate the recognition that (1) the three major concerns from Reviewer 1 have been fully

addressed and (2) removal of the orthokinesis section was appropriate. We also thank the reviewer for recommending expedited publication contingent on the satisfaction of Reviewers 2 and 3. We are grateful for the reviewer's time and positive recommendation.

1. Variability and heterogeneous SRD-1 expression:

The authors attributed much of the variability observed in the study (e.g., Lines 242, 274, and 320) to the partial penetrance of SRD-1 expression in only ~40% of PHD neurons, as reported by the Murphy group. It is not clear to me whether the positive expression of a transgene (such as Chrimson) driven by the *srd-1* promoter in the various strains used in this study can serve as an indicator of SRD-1 expression. If so, the lack of SRD-1 expression in PHD neurons should not be invoked to explain the variability (I would assume that only worms with positive marker expression in PHD neurons were selected for experiments). If not, is there a way to determine SRD-1 expression directly—for instance, by fluorescently tagging the SRD-1 protein and comparing PHD neuron function with or without SRD-1? This approach would be useful for specifically testing SRD-1 function in PHD neurons in future studies. In addition, if SRD-1's function in PHD is important for mate-finding behavior, it would be interesting to discuss how such heterogeneous expression could be maintained evolutionarily.

Reply: We appreciate this thoughtful point and have revised the manuscript accordingly. In the original text we referenced ~40% PHD positivity for *srd-1* from RNA-seq. On re-examination, we note that single-cell/single-nucleus RNA-seq can under-detect low or intermittent transcripts due to technical dropouts and limited sensitivity, particularly for sparsely expressed GPCRs. We now remove the claim that variability arises largely from partial penetrance. We added citations on scRNA-seq detection limits and dropout/zero-inflation modeling. Also, it is very important to mention that the integrated both *srd-1P::cGAL* × UAS::GFP and split-cGAL (*lin-48dP::NLS::cGAL[DBD]*; *srd-1P::NLS::cGAL[AD]* × UAS::GFP) reporter line exhibited uniform PHD labeling (~100%) across assayed animals.

PHD neurons remained active for ≥ 9.5 min with a slow decay (Fig. 5E). Across animals, we observed two response types: fast responders (62.5% of males) showed an immediate, sustained increase at pheromone onset, whereas slow responders (37.5%) showed a ~2.5-min delay and a ~40% lower peak $\Delta F/F$ (Fig. 5E, 0–250 s; Fig. 5D, 0–600 s). Two non-exclusive explanations could account for this heterogeneity. First, molecular variability: single-cell datasets report variation in *srd-1* abundance in PHD. Although our two cGAL-UAS reporter lines label PHD uniformly (~100%), the GAL-UAS system amplifies weak expression and can mask moderate differences in receptor level; thus, the heterogeneity is likely quantitative (more vs. less SRD-1) rather than all-or-none presence/absence. Second, circuit gating: PHD receives postsynaptic inputs from multiple neurons; tonic or state-dependent inhibition from these partners could delay onset and reduce amplitude in some animals. We therefore interpret the fast/slow classes as reflecting threshold-sensing in PHD modulated by receptor dosage and/or inhibitory tone. Distinguishing these mechanisms will require targeted experiments (e.g., *srd-1* dosage manipulation and selective silencing of candidate presynaptic inputs during PHD imaging). We add this to discussion as well.

Page 3: “RNA-seq indicates *srd-1* is detected in ~40% of PHD neurons under standard conditions; however, our integrated *srd-1P*-driven cGAL with UAS::GFP reporter shows uniform PHD labeling (near 100%).”

Discussion, Page 12: “PHD neurons remained active for ≥ 9.5 min with a slow decay (Figure 5E). Across animals, responses segregated into two classes: fast responders (62.5% of males) showed an immediate, sustained increase at pheromone onset, whereas slow responders (37.5%) exhibited a ~2.5-min delay and ~40% lower peak $\Delta F/F$. Two non-exclusive mechanisms could underlie this heterogeneity. (1) Molecular variability:

*single-cell datasets report variation in *srd-1* abundance in PHD. Although both of our *cGAL-UAS* reporter lines labeled PHD uniformly (~100%), *cGAL-UAS* amplification can mask modest expression differences; thus, the heterogeneity is likely quantitative (more vs. less *SRD-1*) rather than all-or-none. Moreover, single-cell/single-nucleus RNA-seq can under-detect low or intermittent transcripts due to limited capture efficiency and technical dropouts/zero inflation—issues known to disproportionately affect sparsely expressed GPCRs (Qiu, 2020; Ziegenhain et al., 2017; Jiang et al., 2022; Bakken et al., 2018; Ehrlich et al., 2018). (2) Circuit gating: PHD receives postsynaptic inputs from multiple neurons; state-dependent inhibition from these partners could delay onset and reduce amplitude in some animals. We therefore interpret the fast/slow classes as reflecting threshold-sensing in PHD modulated by receptor dosage and/or inhibitory factors. Discriminating between these mechanisms will require targeted experiments (e.g., *srd-1* dosage manipulations and selective silencing of candidate presynaptic partners during PHD imaging)."*

2. Spatial separation of AWA and PHD neurons on chemotaxis:

I am not entirely convinced that the spatial separation of these two neurons—in the head and tail, respectively—has no impact on chemotaxis, as the authors argue but this has not been tested. When a worm crawls toward a pheromone source in a gradient, the head neuron AWA would experience a higher pheromone concentration than the tail neuron PHD, and vice versa when crawling away. According to this paper, where AWA and PHD neurons have distinct concentration thresholds, I can't help wondering if the following could be true: when the worm moves in the correct direction, AWA experiences stronger stimulation than PHD, AWA is suppressed, and the worm continues moving forward; conversely, when the worm moves in the wrong direction, AWA receives weaker stimulation than PHD, AWA is activated, and the worm reorients. Although the authors argue that the distance between the head and tail (~1 mm) is too small to matter, this likely depends on how steep the gradient is—particularly when the worm is near the female, where the gradient may be sharp enough to influence the “final sprint.” It would be interesting to know whether the mathematical model could test this hypothesis, or whether it could be experimentally examined by exposing AWA and PHD to different pheromone concentrations using the microfluidic setup shown in Figure 6A and monitoring AWA activity.

Reply: Thank you for discussing this point. Indeed, a more quantitative result will help rationalize our assumption that the concentration difference sensed by AWA (we denote as $C(\text{head})$) and PHD (we denote as $C(\text{tail})$) is not significant and the choice of ignoring it.

We now include a figure in the SI (Fig.S3C) to show the distribution of the relative difference in concentration sensed by head and tail, i.e., $|C(\text{head})-C(\text{tail})|/C(\text{body})$. We average this quantity over (1) the entire plate space, (2) the entire simulation time, and (3) the worm orientation in $(0, 2\pi)$. As shown in this figure, this distribution is concentrated near 0 (notice the left y-axis for the probability density function is in log-scale). The 99% quantile for this distribution is 0.1056, which means that worms will have a $|C(\text{head})-C(\text{tail})|/C(\text{body}) < 0.1056$ for more than 99% of the space-time. A closer examination shows that the larger values for $|C(\text{head})-C(\text{tail})|/C(\text{body})$ are mostly associated with the edge of the plate, where C itself is very small and taking the ratio becomes noisy.

In the model, we included noise for Q_H (head confidence) and Q_T (tail confidence), and we tested with relative noise magnitude 0.01~0.1, showing not much difference in the results. If $|C(\text{head})-C(\text{tail})|/C(\text{body})$ is similar or weaker than the noise amplitude, we do not expect this head-tail concentration difference to be significantly informative for the worm.

Figure S3 figure legend, Page 42: “(C) Distribution of concentration difference between head and tail sensory neurons, normalized by the concentration of body center and averaged over all spatial locations on the plate,

the entire simulation time, and worm orientation. Yellow bars correspond to the left y-axis, showing the empirical probability density distribution (EPDF). We show the y-axis in log-scale to assist visualization since the EPDF is strongly concentrated near 0. The red line corresponds to the right y-axis, showing the empirical cumulative distribution function (ECDF). The 99% quantile is 0.1056.”

Discussion, Page 14: *“To assess whether spatial separation between AWA (head) and PHD (tail) produces a concentration disparity in what they sense, we quantified the relative difference $|C(\text{head}) - C(\text{tail})| / C(\text{body})$ (C , concentration) across our concentration field simulated arenas (Fig. S3C). Specifically, we computed this metric and then averaged over (1) the full plate area, (2) the entire simulation duration, and (3) all worm orientations $\theta \in [0, 2\pi]$. The resulting empirical distribution is strongly concentrated near zero (probability density shown on a log scale), with a 99th percentile of 0.1056. Thus, for >99% of space-time-orientation, $|C(\text{head}) - C(\text{tail})| / C(\text{body}) < 0.1056$. The rare larger values arise predominantly near plate edges where C is itself very small and the ratio becomes numerically noisy, rather than reflecting meaningful signal gradients. Consistent with this result, our model already incorporates stochasticity in head and tail confidence. Across relative noise magnitudes of 0.01–0.1, model outputs changed minimally. Because the typical head–tail concentration contrast is of the same order or smaller than this noise amplitude, we do not expect it to carry reliable information for navigation under our conditions. These quantitative results justify our simplifying assumption to ignore explicit head–tail concentration differences in the main model without altering the qualitative conclusions.”*

Both points are merely suggestions and should not be taken as conditions for publication. The authors have undergone a lengthy review process and have addressed all previous reviewers’ concerns. It would not be fair to request additional revision from a new reviewer at this stage to cause further delay. If Reviewers 2 and 3 are satisfied with the revisions, I would recommend expedited publication, as this paper provides a significant contribution and will be of broad interest to the field.

Reply: We thank the reviewer for the supportive assessment and for recommending expedited publication.

REVIEWERS' COMMENTS

Reviewer #2 (Remarks to the Author):

The authors have improved the presentation of the findings in the text, figures, and figure legends. There are some concerns that remain regarding methodology and these concerns bear on the rigor of the study and its presentation:

1) Gellan Gum — The authors claim that this material by itself creates a "penetrable medium" and, by inference that Agar does not. The authors have added text and citations in support of this claim, but simply citing these other studies is not convincing. Why? Because it is not only the material itself, but also the concentration of the material. Fryer et al use 2.5% (w/v) gellan gum and do not report that this medium is penetrable. By contrast, Wan and Sternberg use 0.3% (w/v) gellan gum to generate a droplet in which single worms can thrive. To avoid misleading others in the field about the performance of gellan gum as an alternative growth medium, please revise the methods to be more precise.

Reply: Thank you for raising this point. We have revised the Methods to be explicit that “penetrable” is a property of the low gellan concentration and resulting mechanics, not of gellan gum per se. In our experiments we used 0.3% (w/v) gellan gum (Gelrite) in M9 buffer on plates, under which worms can enter and locomote within the gel volume. We now avoid implying that gellan gum at higher concentrations (e.g., ~2–3% w/v used in other applications) is similarly penetrable, and we clarify that those stiffer formulations are not expected to permit volumetric penetration.

Line 738-747: *“We prepared plates containing 0.3% (w/v) gellan gum in M9 buffer to generate a soft hydrogel substrate in which worms could enter and locomote within the gel volume, rather than being confined to surface crawling. This ‘penetrable’ behavior is not an intrinsic property of gellan gum, but depends strongly on gel concentration and the resulting mechanical properties. Higher gellan gum concentrations commonly used to cast firm gels (e.g., ~2–3% w/v) produce substantially stiffer matrices and are not expected to permit volumetric penetration. Accordingly, our statement refers specifically to the 0.3% (w/v) gellan-in-M9 plates used here, which we selected as the lowest concentration that maintained plate integrity while supporting robust 3D locomotion [123]. Gellan/Gelrite has been adopted for chemotaxis arenas in high-throughput screening and behavioral assays [124,125]. Penetrable gels and 3D habitats are known to elicit burrowing and other naturalistic behaviors [126-128].”*

2) Throughout the manuscript the authors use "velocity" to describe the rate of animal movement. Strictly speaking, velocity is a vector quantity consisting of both direction and rate. Please consider revising the language to use the term "speed" in its place.

Reply: We have revised the manuscript and figures to replace “velocity” with “speed” throughout.

Reviewer #4 (Remarks to the Author):

The revised manuscript has addressed all the points raised by me and the other reviewers, and it is much improved in both clarity and rationale. I especially appreciate the addition of Fig. S3C, which quantitatively addresses my previous concern. Only one very minor point: the “red line” in Fig. S3C does not appear red, at least in the version of the PDF I received.

Reply: Thank you for these very helpful suggestions and feedback. We appreciate the opportunity to address this point quantitatively. It helped make the discussion more interesting. We have revised the legend/description to refer to the line as orange (rather than red).